# Understanding the effect of spatially separated Cu and acid sites in zeolite catalysts on oxidation of methane

Peipei Xiao[1], Yong Wang[1], Lizhuo Wang[2], Hiroto Toyoda[1], Kengo Nakamura[1], Samya Bekhti [1], Yao Lu[1], Jun Huang [2], Hermann Gies [1,3] & Toshiyuki Yokoi [1,4] ✉

Unraveling the effect of spatially separated bifunctional sites on catalytic reactions is significant yet challenging. In this report, we investigate the role of spatial separation on the oxidation of methane in a series of Cu-exchanged aluminosilicate zeolites. Regulation of the bifunctional sites is done either through studying a physical mixture of Cu-exchanged zeolites and acidic zeolites or by systematically varying the Cu and acid density within a family of zeolite materials. We show that separated Cu and acid sites are beneficial for the formation of hydrocarbons while high-density Cu sites, which are closer together, facilitate the production of $CO_2$. By contrast, a balance of the spatial separation of Cu and acid sites shows more favorable formation of methanol. This work will further guide approaches to methane oxidation to methanol and open an avenue for promoting hydrocarbon synthesis using methanol as an intermediate.

With the gradual depletion of crude oil, growing demand for chemicals, and cumulative greenhouse gas emissions, the catalytic conversion of $CH_4$ and $N_2O$ to value-added chemicals is an urgent and significant issue[1–4]. Traditionally, the $CH_4$ resource utilization is first through steam methane reforming (SMR) process to produce syngas[5,6], then Fischer-Tropsch (FT) synthesis to form methanol[7–9], and followed by the methanol-to-olefin (MTO) conversion to generate olefins[10,11]. However, the long and energy-intensive reaction process needs to be simplified. In addition, the cleaning of $N_2O$ ($N_2O \rightarrow N_2 + O_2$) relies on high temperature or catalytic thermal decomposition[12]. Direct catalytic conversion of $CH_4$ and $N_2O$ to methanol or other value-added chemicals is possible to alleviate the above-mentioned problems[13,14]. Actually, the direct conversion of syngas to hydrocarbons over metal-oxide/zeolite bifunctional catalysts via a tandem reaction coupling approach is the focus of recent research and has achieved breakthrough progress[15]. The strategy was designed to implement the short process of merging two steps into one[4]. In addition, direct conversion of methane to valuable chemicals is able to be realized via oxidative coupling of methane (OCM)[16,17], methane dehydroaromatization (MDA)[18–20], and the newly emerging strategy of methane conversion to olefins, aromatics, and hydrogen (MTOAH)[21,22]. However, the three strategies require a quite high temperature (å 700 ºC) and the products distribution is limited in ethylene, ethane, and benzene[21]. In the recent work of Sushkevich et al. oxidative methane C – C coupling over copper-exchanged zeolites was reported[23]. However, the coupling products are light paraffins, not light olefins.

To our knowledge, the strategy of direct conversion of methane to hydrocarbons (DMTH) over metal-containing zeolite bifunctional catalysts via a tandem reaction coupling approach has not been sufficiently studied. In the past decades, the focus of direct oxidation of methane to methanol was on the yield and selectivity of methanol[13,24]. In order to improve methanol selectivity, lower temperatures (200–300 ºC) were regularly used to prevent methanol to hydrocarbons (MTH) reaction and the over-oxidation of methanol to CO and $CO_2$[13,14,24–26]. Therefore, hydrocarbons as the secondary reaction products of methane oxidation have received little attention. Recent work

[1]Institute of Innovative Research, Tokyo Institute of Technology, 4259 Nagatsuta, Midori-ku, Yokohama 226-8501, Japan. [2]School of Chemical and Biomolecular Engineering, The University of Sydney, Sydney, NSW 2006, Australia. [3]Institute of Geology, Mineralogy und Geophysics, Ruhr-University Bochum, Bochum 44780, Germany. [4]iPEACE223 Inc. Konwa Building, 1–12-22 Tsukiji, Chuo-ku, Tokyo 104-0045, Japan. ✉e-mail: yokoi@cat.res.titech.ac.jp

of direct oxidation of methane to methanol reported by Xu et al. has detected dimethyl ether (DME) and hydrocarbons at 320 °C, however, the coke selectivity was up to 80% and the selectivity of hydrocarbon has not been fully researched due to the use of large pore Cu/BEA zeolite[27]. Very recently, we performed direct oxidation of methane to methanol (DMTM) reaction at 350 °C on the small pore Cu/AEI zeolite and observed hydrocarbons being converted from methanol on acid sites of Cu/AEI zeolite, where AEI is the abbreviation of Aluminophosphate-eighteen and the aluminosilicate AEI-type zeolite is well-known as SSZ-39[28–30]. Simultaneously, we noticed that some Cu/AEI zeolites contained a considerable amount of Brønsted Acid Sites (BAS), however, no hydrocarbons were observed. Thus, other factors affecting the generation of hydrocarbons should be considered.

Inspired by the achievements in conversion of syngas to olefins via methanol/DME intermediates on metal-oxide/zeolite catalysts[31], the effect of spatially separated $Cu^{2+}$ and BAS in the Cu-exchanged AEI zeolite on the reaction performance of direct oxidation of methane was studied in this work. Noteworthy, the Ångstrom and nanoscale intimacy are quite difficult to describe and investigate. A study by Martens and coworkers uncovered that variation in the distance between Pt nanoparticles and BAS in the association of Pt-$Al_2O_3$/Y zeolite and Pt-Y zeolite/$Al_2O_3$ achieved different product selectivity in n-Decane hydrocracking reactions[32]. Only Pt was observable in the HAADF-STEM images, however, visualization of Pt and acid sites over spatial distance remained elusive. Similar research has been reported several times by de Jong and coworkers[33,34]. In addition, Deng and coworkers adopted the $^1H$–$^{67}Zn$ double-resonance solid-state NMR technique to detect the spatial proximity/interaction between $Zn^{2+}$ and $H^+$ from BAS in Ångstrom scale[35]. However, it was still difficult to specifically quantify the spatially separated metal ions and BAS in a large-scale range.

Herein, we presented the simple strategy to control the spatially separated Cu and BAS by ion exchange and physical mixture and thus to investigate its influence on the reaction performance of direct oxidation of methane. When AEI zeolite was ion-exchanged with different concentrations of $Cu(NO_3)_2$ solution, the density of Cu and BAS were calculated based on the chemical composition and cell parameters. When Cu/AEI was physically mixed with acidic zeolite in different integration manners, the spatially separated Cu and BAS were modeled. The reaction performance and spatial distance were well associated. These results would guide reasonable design catalysts and reaction conditions according to the target products.

## Results and discussion

### Effect of the spatially separated Cu and acid sites in Cu/AEI zeolite

The basic characterization results of xCu/AEI zeolites have been presented (Supplementary Figs. 1-4). Since no peak at 35.6 and 38.8° belonging to aggregated CuO nanoparticles was observed, the highly dispersed Cu species were obtained (Supplementary Fig. 1)[36]. The UV-vis spectra, HRTEM images, HADDF-STEM images, and EDS mapping images further confirmed the highly dispersed feature of Cu species in xCu/AEI samples (Fig. 1a–d) and Supplementary Figs. 5–8[37].

The xCu/AEI zeolite catalysts with varying Cu content (Supplementary Figs. 8–12 and Supplementary Tables 1–3) were used to perform the intimacy effect between Cu and BAS on the reaction performance of DMTM (Supplementary Fig. 13). Although Al and Cu can be visualized through EDS mapping images (Fig. 1a–d, Supplementary Figs. 6–8), it is still full of challenges to quantify the variation in spatially separated BAS and Cu. The reaction performance is shown in Fig. 1e–h, MeOH was produced at the initial stage (TOS = 0.16 h) for all the samples and a close MeOH formation rate was obtained (ca. 18-25 μmol.g$^{-1}$.min$^{-1}$). However, as the reaction proceeded, MeOH was massively and rapidly converted to hydrocarbons on the catalysts with low Cu content within 5 h (Fig. 1e, f). The massive formation of

hydrocarbons on 1Cu/AEI zeolite led to a severe decrease in the formation rate and selectivity of MeOH (from 25 to 1.3 μmol·g$^{-1}$·min$^{-1}$ and from 25 to 2%, respectively) (Fig. 1e). After a few hours of reaction, the coke was deposited on the acid sites, resulting in the inactivation of acid sites, which were confirmed by comparison of the results of $NH_3$-TPD and TG-DTA for the fresh and spent zeolites (Supplementary Figs. 14, 15). In addition, another reason that caused the gradually declined hydrocarbon formation rate was dealumination in the process of activation and reaction, which has been proven by comparison of the $^{27}Al$ MAS NMR curves of the fresh and spent zeolites (Supplementary Fig. 16). It is noteworthy that the activity of Cu sites was still maintained, thus the productivity of MeOH was able to recover (Supplementary Figs. 14c, d). As for hydrocarbons, $C_2^=$ was formed first with the highest selectivity in hydrocarbons. With the Cu content increasing and the corresponding acid amount decreasing (5Cu/AEI), the formation time of hydrocarbons was prolonged in comparison with 1Cu/AEI, indicating that the transformation of MeOH from the Cu sites to the acid sites required a longer time possibly due to the farther separation (Fig. 1f). On the contrary, almost no hydrocarbons were observed on the catalysts with more Cu content and less acid sites (Fig. 1g, h). We considered that the significant difference in the formation of hydrocarbons was due to the reduced amount of BAS (Supplementary Table 2), or the farther and more unattainable spatially separated Cu and the acid sites (Supplementary Fig. 17). 50 Cu/AEI and 500 Cu/AEI showed different $CO_2$ selectivity compared to 1Cu/AEI and 5Cu/AEI, signifying that the excessive Cu amount led to over-oxidation[23].

Since it was agreed that the peak centered ca. 475 °C in the $NH_3$-TPD curve was due to $NH_3$ strongly adsorbed on the BAS, this peak was used to inspect the impact of BAS content or strong acid amount on the MTH reaction[38]. The impact of Cu density (Supplementary Table 1) and strong acid amount (Supplementary Table 2) on the product selectivity of direct oxidation of methane at TOS = 1.16 h was visualized (Fig. 1i–k). Cu density was inversely proportional to strong acid amount (Fig. 1i), due to the Cu ions replacing H of Si(OH)Al. $CO_2$ selectivity proportionally grew to Cu density, while the opposite trend was observed for hydrocarbon selectivity. In addition, the selectivity of (MeOH+DME) displayed the hill shape along with Cu density (Fig. 1j). Therefore, the hydrocarbon selectivity increased along with the strong acid amount (Fig. 1k), the selectivity of CO and $CO_2$ showed opposite trends, and that of MeOH and DME peaked at the medium amount of strong acid sites.

Herein, we considered schematic diagrams of the spatially separated Cu and BAS for 1Cu/AEI and 500Cu/AEI (Supplementary Figs. 17a, b). Based on the cell parameters and the chemical composition, and most importantly, assuming that all elements were uniformly distributed and ignoring Al zoning, a unit cell (a = 13.7 Å, b = 12.6 Å, c = 18.5 Å) of 1Cu/AEI and 500Cu/AEI contained 0.24 and 1.44 Cu atoms (Supplementary Table 1), respectively. Namely, 4 unit cells shared ca. 1 Cu atom in 1Cu/AEI and ca. 6 Cu atoms in 500Cu/AEI. Simply assume that a single Cu corresponds to a single Al, and Cu pairs correspond to Al pairs. When 4 unit cells shared ca. 1 Cu atom, 24 Al atoms, and thus 23 H atoms from BAS, the spatially separated Cu and acid sites would be relatively close in Ångstrom scale. Consequently, it was quite possible to transport the intermediate MeOH from Cu sites to acid sites and convert it to hydrocarbons. However, when 4 unit cells shared ca. 6 Cu atoms, 24 Al atoms, and thus 18 H atoms from bridging hydroxyl. The possibility that MeOH produced on Cu sites was transported to acid sites was reduced. Therefore, the maximum selectivity of hydrocarbons declined from 95% for 1Cu/AEI to 0.3% for 500Cu/AEI. Furthermore, the increased Cu density resulted in the closely located Cu and Cu sites, which was visible in the representative and comparison HAADF-STEM images of 1Cu/AEI and 500Cu/AEI (Supplementary Fig. 7), and thus the methanol intermediate was over-oxidized to CO and $CO_2$[23].

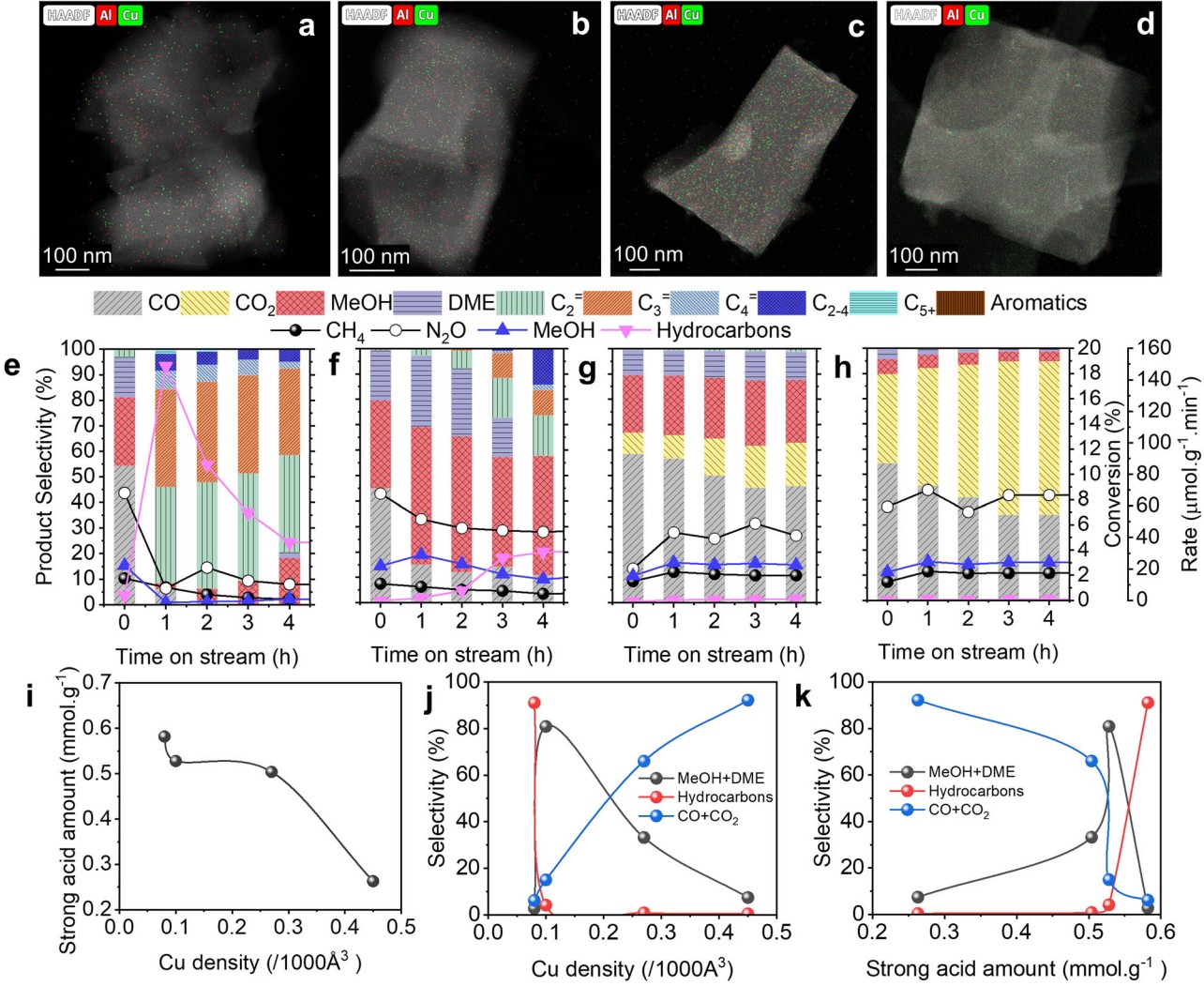

**Fig. 1 | Effects of Cu density and acid amount on reaction performance of direct oxidation of methane.** EDS elemental mapping images of overlay Al and Cu for (**a**) 1Cu/AEI, (**b**) 5Cu/AEI, (**c**) 50Cu/AEI, (**d**) 500Cu/AEI zeolite catalysts. Reaction performance of (**e**) 1Cu/AEI, (**f**) 5Cu/AEI, (**g**) 50Cu/AEI, (**h**) 500Cu/AEI at 350 °C. Reaction conditions: 100 mg catalyst, $CH_4/N_2O/H_2O/Ar = 10/10/2/3$ ml·min$^{-1}$, WHSV = 15000 ml·g$^{-1}$·h$^{-1}$. $r_{Hydrocarbons} = 2(r_{C2H4} + r_{C2H6}) + 3(r_{C3H6} + r_{C3H8}) + 4(r_{C4H8} + r_{C4H10}) + 5r_{C4+}$. **i** The relation of Cu density with the strong acid amount; (**j**) product selectivity as a function of Cu density (TOS = 1.16 h); (**k**) product selectivity as a function of the strong acid amount (TOS = 1.16 h).

To verify our hypothesis, the diffusion velocity was adjusted by varying the flow rate and catalyst amount of 5Cu/AEI and 500Cu/AEI zeolites (Supplementary Figs. 18, 19). When the total flow rate was doubled by raising Ar flow rate from 3 to 28 ml·min$^{-1}$, which meant that the weight hourly space velocity (WHSV) augmented from 15,000 to 30,000 ml·g$^{-1}$·h$^{-1}$, the peak selectivity of hydrocarbons for 5Cu/AEI was enlarged from *ca.* 45 to 65% and the massive formation time of hydrocarbons was advanced from 4.16 to 2.16 h (Supplementary Fig. 18a, b). While for 500Cu/AEI, growing WHSV improved the selectivity of $CO_2$ and DME (Supplementary Fig. 18c, d). These results suggested that intensification of space velocity was able to improve the mass transportation between Cu and acid sites as well as between Cu and Cu sites.

Moreover, similar results were obtained by regulating the catalyst amount from 25 to 200 mg. For 5Cu/AEI, the peak selectivity of hydrocarbons was amplified from *ca.* 3 to 75%, and the massive formation time of hydrocarbons was advanced to 1.16 h (Supplementary Figs. 19a, b). Because MeOH is first formed on the Cu sites of Cu/AEI zeolite in the front bed, and then, the produced MeOH goes through the acid sites of Cu/AEI in the back bed and is converted to hydrocarbons[30]. In the case of 500Cu/AEI, more catalyst amount

resulted in higher $CO_2$ selectivity due to the dominant role of Cu sites of 500Cu/AEI in the back bed (Supplementary Fig. 19c, d).

Subsequently, the acid density was adjusted by calcination at 550–850 °C after Cu exchange to guarantee a similar total Cu and Al content[39]. Aggregated Cu clusters were not observed even after calcination at 850 °C (Supplementary Figs. 20–26). The strong acid amount decreased with the calcination temperature growing from 550 to 850 °C (Supplementary Fig. 26 and Supplementary Table 2). As for the reaction performance (Fig. 2a–c), the selectivity of total hydrocarbons peaked *ca.* 40% at 4.16 h on 5Cu/AEI-550 (*i.e.* 5Cu/AEI zeolite). Continuing to diminish the strong acid amount, the selectivity of total hydrocarbons peaked at *ca.* 20 % at 6.16 h on 5Cu/AEI-750 and was less than 2% within 10.16 h on 5Cu/AEI-850. One should note that the methanol formation rate of 5Cu/AEI-850 amplified and stabled at *ca.* 33 μmol·g$^{-1}$·min$^{-1}$ (1980 μmol·g$^{-1}$·h$^{-1}$) due to the low selectivity of total hydrocarbons, which was the source of carbon deposit. The relationship between the strong acid amount and product formation rate was depicted in Fig. 2d[40]. Only the formation rate of hydrocarbons was proportional to the strong acid amount, both the formation rate of total products and (MeOH+2*DME) were inversely proportional to the strong acid amount. Considering the mechanism of acid amount

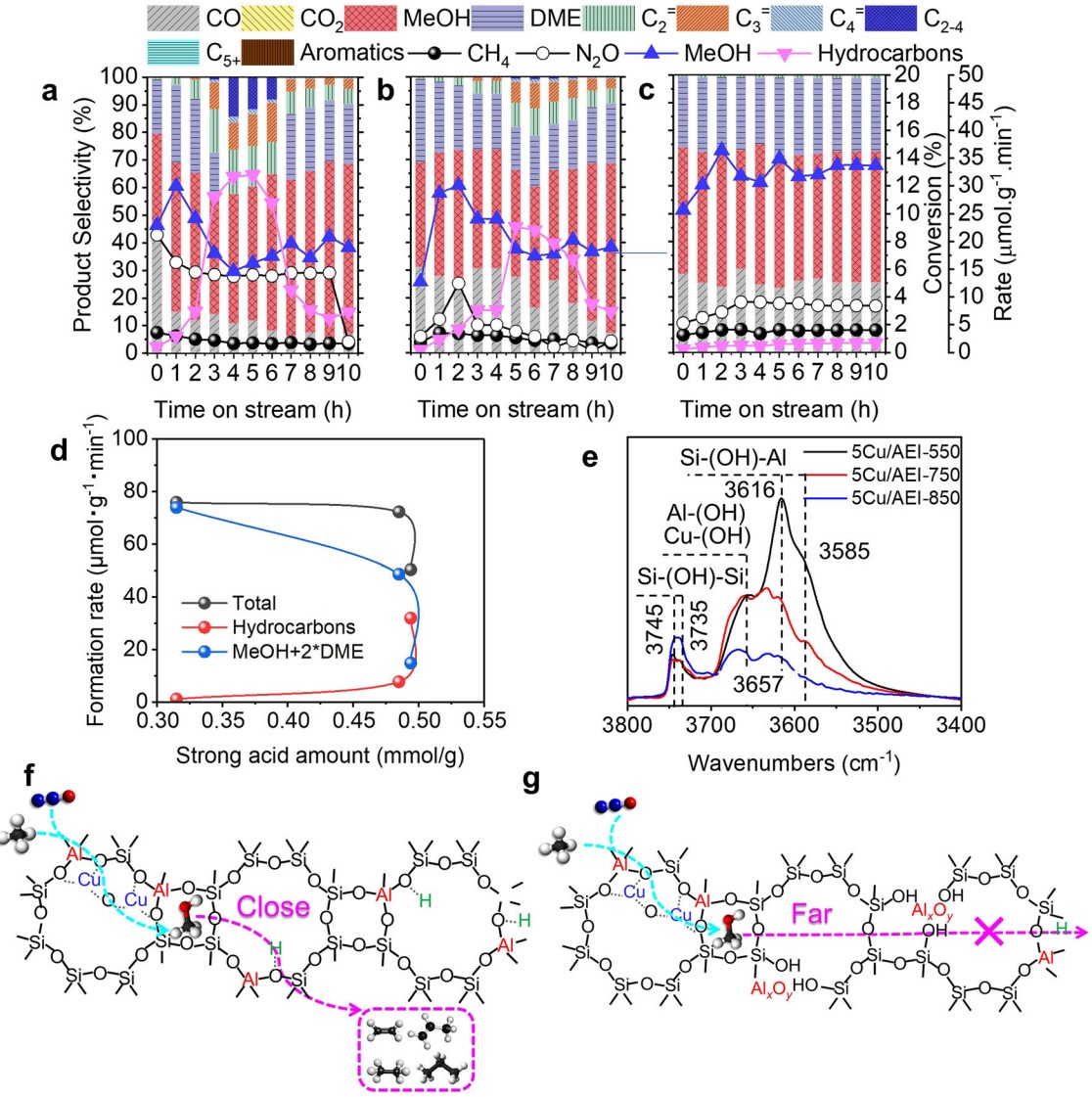

**Fig. 2 | Effects of acid amount on reaction performance of direct oxidation methane.** Stability test of (**a**) 5Cu/AEI-550, (**b**) 5Cu/AEI-750, (**c**) 5Cu/AEI-850 in DMTM reaction at 350 °C. Reaction conditions: 100 mg catalyst, CH₄/N₂O/H₂O/Ar = 10/10/2/3 ml·min⁻¹, WHSV = 15000 ml·g⁻¹·h⁻¹. **d** Product formation rate as a function of strong acid amount at TOS = 4.16 h, $r_{Hydrocarbons} = 2(r_{C2H4} + r_{C2H6}) + 3(r_{C3H6} + r_{C3H8}) + 4(r_{C4H8} + r_{C4H10}) + 5r_{C4+}$,

$r_{Total} = r_{CO} + r_{CO2} + r_{MeOH} + 2(r_{DME} + r_{C2H4} + r_{C2H6}) + 3(r_{C3H6} + r_{C3H8}) + 4(r_{C4H8} + r_{C4H10}) + 5r_{C4+}$. **e** FTIR spectra in the OH stretching region of 5Cu/AEI-*t* samples collected at −120 °C after activation at 500 °C in vacuum for 1 h. Schematic depictions of (**f**) closely (5Cu/AEI-550) and (**g**) distantly (5Cu/AEI-850) separated Cu and acid sites on the reaction performance of direct oxidation of methane.

reduction by calcination (Fig. 2e), the schematic diagram of the relationship between Cu and BAS was depicted in Fig. 2f, g and Supplementary Fig. 27. It is worth pointing out that cations such as Cu²⁺ and Na⁺ on the extra framework of zeolite can stabilize Al in the framework[41]. Thus, dealumination via high-temperature calcination mainly occurred on Al of Si(OH)Al (Fig. 2f, g). When the sample was calcined at a low temperature (550 °C), most of Al atoms in the framework were reserved. The content of BAS was sufficient and the spatially separated Cu and acid sites was relatively close (Fig. 2f). Therefore, the MeOH and DME intermediates were transported to the acid sites and converted to hydrocarbons (Fig. 2f). When 5Cu/AEI was calcined at higher temperatures (750 and 850 °C), the vacancies and reduced acid content happened because of dealumination (Fig. 2g). Correspondingly, the intimacy between Cu and acid sites weakened, and thus delivery of MeOH from Cu sites to acid sites took longer time or even was difficult to deliver from Cu sites to the farther acid sites (Fig. 2g)[42]. Simultaneously, because the Cu content and Cu states offered negligible changes (Supplementary Figs. 24, 25), few MeOH

and DME were over-oxidized to CO₂ (Fig. 2c). The carbon deposit amount of 5Cu/AEI-*t* decayed from 8.0 to 5.2% with *t* increasing from 550 to 850 °C (Supplementary Fig. 28).

## Effect of the spatially separated Cu/AEI and H-AEI

Since it was difficult to measure the spatial distance between Cu and acid sites in Cu/AEI zeolites, the physical mixture of Cu/AEI zeolite and acidic zeolite was adopted to explore the effect of spatial separation on reaction performance. Here, 50Cu/AEI zeolite catalyst was adopted as the Cu component because hydrocarbons were not generated and the CO₂ selectivity was moderate (Fig. 1g). Meanwhile, H-AEI zeolite was used as the acidic component (Supplementary Table 2). A series of integration manners between 50Cu/AEI and H-AEI zeolite were used to regulate the spatially separated Cu and acid sites from the millimeter to nanometer. Based on the literature, the spatially separated two functional granules was roughly considered as the sum of two radii for the granule stacking and powder-grinding manner[30]. In this study, the same size of pellets (400–600 μm) and particles

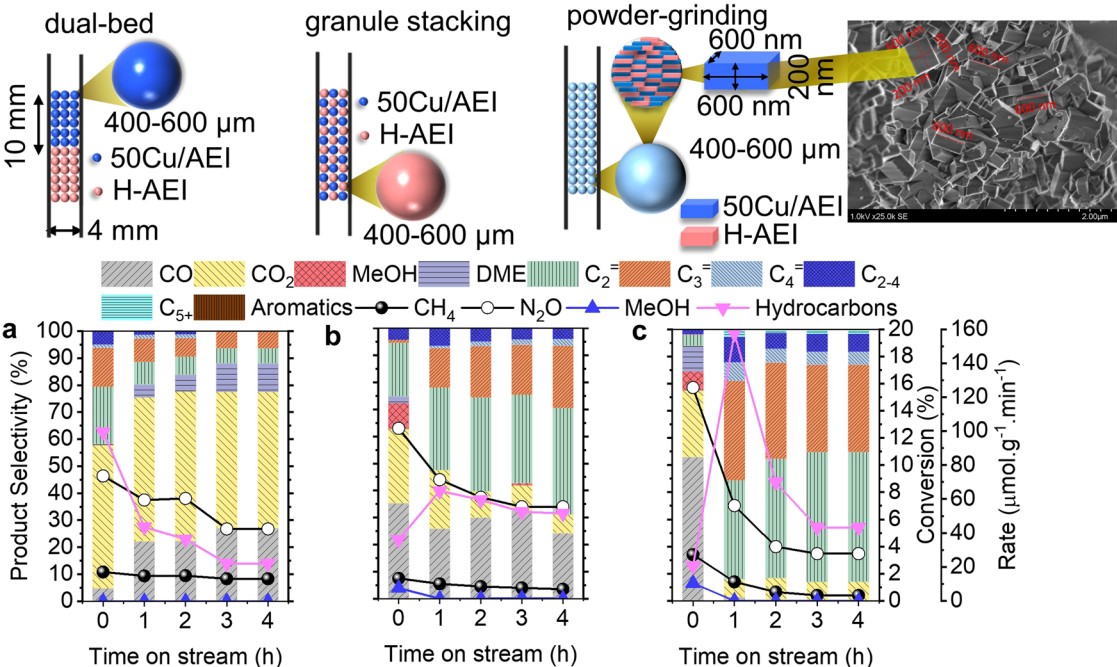

**Fig. 3 | Effects of mixing methods of 50Cu/AEI and H-AEI on reaction performance of direct oxidation of methane. a** Dual bed (50 mg 50Cu/AEI + 50 mg H-AEI); (**b**) granule-stacking of 50 mg 50Cu/AEI and 50 mg H-AEI with size of 400–600 μm; (**c**) grinding-mixing 50% 50Cu/AEI and 50% H-AEI and then compressing to granules with size of 400–600 μm. Reaction conditions: total catalyst mass: 100 mg, 350 °C, $CH_4/N_2O/H_2O/Ar=10/10/2/3$ ml·min$^{-1}$, WHSV = 15,000 ml·g$^{-1}$·h$^{-1}$. $r_{Hydrocarbons} = 2(r_{C2H4} + r_{C2H6}) + 3(r_{C3H6} + r_{C3H8}) + 4(r_{C4H8} + r_{C4H10}) + 5r_{C4+}$.

(200 nm* 600 nm* 600 nm) of Cu/AEI and H-AEI zeolite catalysts were used, and the spatially separated Cu and acid sites was approximately equal to the size of pellets (400–600 μm) and particles (200–600 nm) of AEI zeolite.

The spatially separated Cu species in 50Cu/AEI and acid sites in H-AEI was regulated from millimeter (Fig. 3a), micrometer (Fig. 3b) to nanometer (Fig. 3c) by attempting the dual-bed, granule stacking, and powder-grinding modes. The separation of 50Cu/AEI and H-AEI by quartz wool in one reactor, i.e., the dual-bed model, led to no methanol being observed during the initial 6 h (Fig. 3a), suggesting that all the intermediate MeOH was transferred to hydrocarbons on the acid sites of the second-bed H-AEI zeolite. Meanwhile, the total selectivity of hydrocarbons reduced from 42 to 13% within 6 h. In this model, the formation rate of total hydrocarbons peaked at the beginning (TOS = 0.16 h), signifying that the intermediate MeOH was transported to the acid sites quickly and easily. In addition, the acid sites in this model were easy to inactivate, possibly due to the acidic zeolite located in the second bed. When we stacked granules of 50Cu/AEI and H-AEI in the size of 400-600 μm, the selectivity of total hydrocarbons was improved from 25 to 75% as the reaction proceeded to 6 h (Fig. 3b). The gradually increased selectivity of hydrocarbons implied that the MeOH produced on 50Cu/AEI was converted at the acid sites on H-AEI step-by-step. Furthermore, when the powder-grinding 50% 50Cu/AEI and 50% H-AEI was applied in the oxidation of methane reaction, the maximum selectivity of total hydrocarbons was up to 92% at TOS = 1.16 h (Fig. 3c). Compared to the results of Fig. 3, the selectivity of CO$_2$ was decreased to 24–7% from 58–50% with the spatially separated Cu and acid sites declining. Hence, it has been intuitively proved that the distantly separated Cu and acid sites was beneficial to the production of CO$_2$, while the close separation was favorable to the formation of hydrocarbons.

It was worth pointing out that the product distribution, especially CO and CO$_2$, was highly dependent on the Cu component. The different Cu species on 50Cu/AEI, 500 Cu/AEI, and 5Cu/AEI-850 zeolite catalysts were used as the Cu sites (Supplementary Figs. 9c, d, 25c)

since almost no hydrocarbons were obtained (Figs. 1g, h, 2c). The reaction performance of stacking granules (50Cu/AEI and H-AEI), (500Cu/AEI and H-AEI), and (5Cu/AEI-850 and H-AEI) was compared (Supplementary Fig. 29). In the premise of a similar spatially separated Cu and BAS, *ca.* 27, 48, and 6% CO$_2$ selectivity were obtained on (50Cu/AEI and H-AEI), (500Cu/AEI and H-AEI), and (5Cu/AEI-850 and H-AEI), respectively, due to the different Cu component.

Additionally, AEI and BEA zeolites were adopted as the small and large pore zeolites, respectively, to research the effect of Cu component and spatially separated Cu and BAS (Supplementary Fig. 30a, b). 5Cu/BEA zeolite was prepared using the same method as 5Cu/AEI. Analogous to 5Cu/AEI, the XRD pattern, UV-vis spectrum, HAADF-STEM images and EDS mapping images of 5Cu/BEA indicated that aggregated CuO nanoparticles were not observed (Supplementary Fig. 30c–e). The broad adsorption band at 650-850 nm of 5Cu/BEA was assigned to the d → d transition[43]. Combine with the NO adsorption FTIR spectra of 5Cu/BEA, the Cu$^{2+}$(NO) bands at higher wavenumbers of 1959, 1948, and 1936 cm$^{-1}$ possibly indicated the Cu clusters in 5Cu/BEA zeolite (Supplementary Fig. 30f). The NH$_3$-TPD curves specified that the acid amount of 5Cu/BEA was less than that of 5Cu/AEI due to the higher Si/Al ratio and larger pore size of Beta zeolite (Supplementary Fig. 30g and Supplementary Table 2). The Cu clusters, less acid amount, weaker acid strength, and the distantly separated Cu and BAS on 5Cu/BEA zeolite resulted in *ca.* 75 % CO$_2$ selectivity and no hydrocarbons, while the main product of 5Cu/AEI was MeOH (48 %) (Supplementary Fig. 30h, i). Additionally, the compared characterization and reaction results of 1Cu/AEI and 1Cu/BEA zeolites further confirmed the statement (Supplementary Fig. 31). Therefore, the result reasonably validated our hypothesis that the distantly separated Cu and acid sites, and the closely separated Cu and Cu sites were convenient for the formation of CO$_2$.

## Effect of the acidic zeolite

The reaction performance of different topology structure zeolites as the acidic zeolites, including H-AEI (BASF SE), H-MFI (Japan Reference

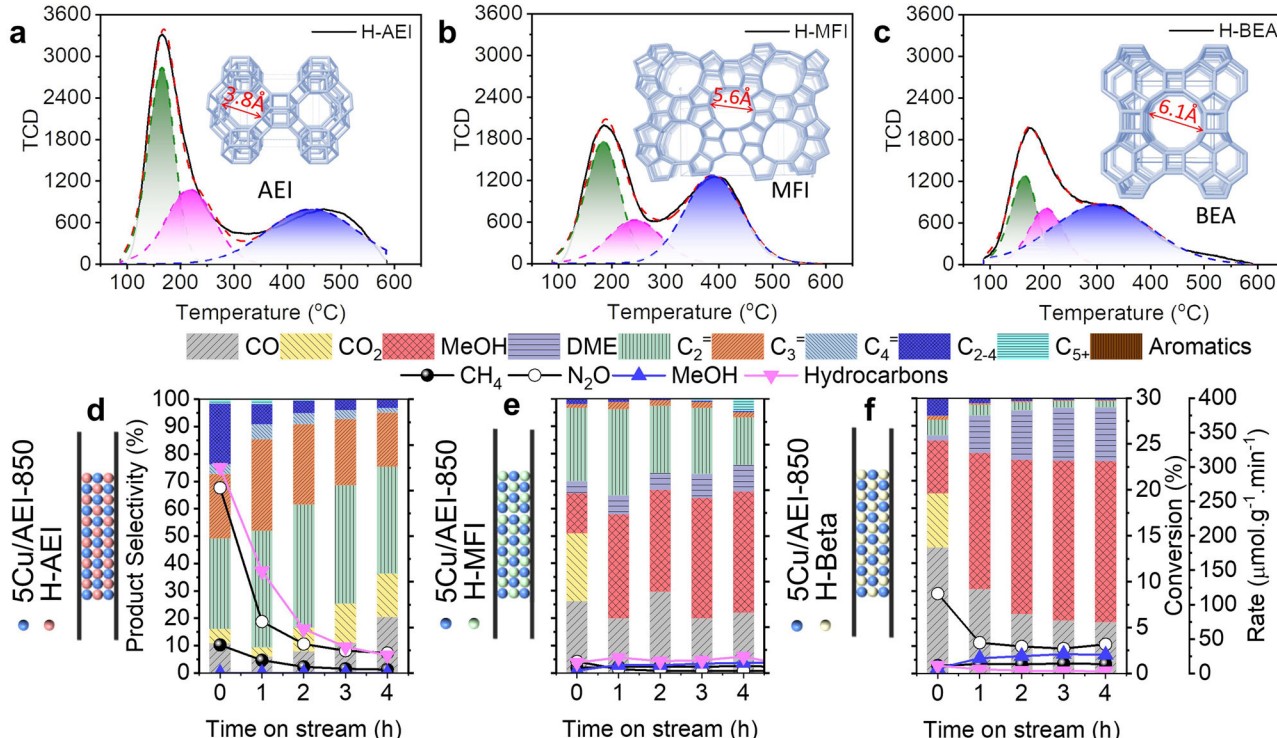

**Fig. 4 | Effects of the acidic zeolite structure on reaction performance of direct oxidation of methane.** Deconvolution of NH$_3$-TPD curves of (**a**) H-AEI, (**b**) H-MFI, (**c**) H-BEA zeolites. Reaction performance of 5Cu/AEI-850 granular stacking with (**d**) H-AEI, (**e**) H-MFI, (**f**) H-BEA in the direct oxidation of methane. Reaction conditions:

50 mg 5Cu/AEI-850 and 50 mg H-zeolite with the granule size of 400–600 μm, 350 °C, CH$_4$/N$_2$O/H$_2$O/Ar=10/10/2/3 ml·min$^{-1}$, WHSV = 15,000 ml·g$^{-1}$·h$^{-1}$. $r_{Hydrocarbons} = 2(r_{C2H4} + r_{C2H6}) + 3(r_{C3H6} + r_{C3H8}) + 4(r_{C4H8} + r_{C4H10}) + 5r_{C4+}$.

Catalyst, JRC-Z5-30NH$_4$), and H-BEA (Zeolyst International, CP814E*) (Supplementary Table 3 and Supplementary Fig. 32), and 5Cu/AEI-850 as the Cu component, in the granule-stacking model with the size of 400–600 μm were compared in Fig. 4. First of all, the product distribution was different. For the small pore zeolite AEI, the selectivity of hydrocarbons was up to 90%, and neither MeOH nor DME was remained as the intermediates (Fig. 4d). While for the medium (MFI) and large (BEA) pore zeolites, the selectivity of hydrocarbons dropped to 30 and 14 % (Fig. 4e, f), respectively. Meanwhile, *ca.* 60 and 75% selectivity of (MeOH+DME) were left through MFI and BEA, respectively, indicating that both topology structure and acidity of acidic zeolite played a vital role. The remaining MeOH and DME were the apparent indication of the insufficient acid content, which was confirmed by the NH$_3$-TPD results (Fig. 4b, c and Supplementary Table 2), and an indication of inaccessible acid sites. Moreover, another possible reason was the spatially separated Cu and acid sites. As previously mentioned, medium and large pore-size zeolites meant a distantly separated acid site. When medium and large pore-size zeolites were used as the acid sites, the spatially separated Cu sites in 5Cu/AEI-850 zeolite and acid sites in H-MFI or H-BEA zeolites became farther, resulting in the lower selectivity of hydrocarbons. Meanwhile, MeOH and DME that have not been promptly converted were detected by GC-FID as the final products.

In our previous work, CHA zeolite was used in DMTM reaction and ethylene (C$_2^=$) was observed as the main hydrocarbon component, and we ascribed to the "hydrocarbon pool (HCP)" mechanism[44], which was proposed and accepted in the MTO or MTH process[10]. There was no dispute under that circumstance since the C$_2^=$ selectivity of CHA zeolites was reported higher than other olefins in the MTH reaction due to the cage effect[45–47]. Similarly, in the publication of Su et al. the mechanism of syngas to light olefins on ZnCrOx/AlPO-18 catalyst was also ascribed to the "HCP" mechanism[14]. However, in this study, we found that C$_2^=$ was always

the main contributor among the hydrocarbons regardless of the acidic properties and the spatially separated Cu and acid sites (Supplementary Figs. 33–35). It was different from the MTH reaction result taking H-AEI zeolite as an example (Supplementary Fig. 36) and also different from the reports that propylene (C$_3^=$) was the main hydrocarbon product[39,40,45,48]. To shed light on the doubt, we proposed the possible reaction route in Fig. 5, where MeOH is produced on the Cu sites as the initial product, and two MeOH molecules would form DME on the acid sites through the dehydration reaction[49]. In this study, the production of hydrocarbons was considerably dependent on the transportation of MeOH to the acid sites. Thus, both the closely located Cu and acid sites (Figs. 1–3) and the accelerated mass transfer (Supplementary Fig. 18) were able to realize the high selectivity and earlier formation of hydrocarbons. When MeOH is promptly transported to the acid sites and C$_n$H$_{(2n+1)}$ intermediates are not released from the ((C$_n$H$_{(2n+1)}$)···O)- sites, it is possible to add another methyl group and then yield to C$_m$H$_{2m}$, where $m$ is equal to $n + 1$ ($n \geq 1$). The hydrocarbons distribution of 5Cu/AEI (Supplementary Fig. 37) exposed that various alkenes and alkanes were gradually produced then disappeared. In addition, the hydrocarbons selectivity was increased with the flow rate growing (Supplementary Fig. 18b). Both the two phenomena were able to verify the conjecture. In fact, the methoxymethyl cation mechanism and methane−formaldehyde mechanism[44,50] were proposed as competitive mechanisms and important complements to the "HCP" mechanism in MTH reaction[51]. In this work, formaldehyde (HCOH) was not observed due to the limitation of the high reaction temperature (350 °C) and analytical instrument. Thus, there is not enough evidence for methane−formaldehyde mechanism. On the other hand, the formative sequence of hydrocarbons in this study was consistent with the methoxymethyl cation mechanism, which has been reported as the initial C-C bond formation mechanism in MTH reaction[50,52,53] and syngas to olefins reaction[54,55]. Furthermore,

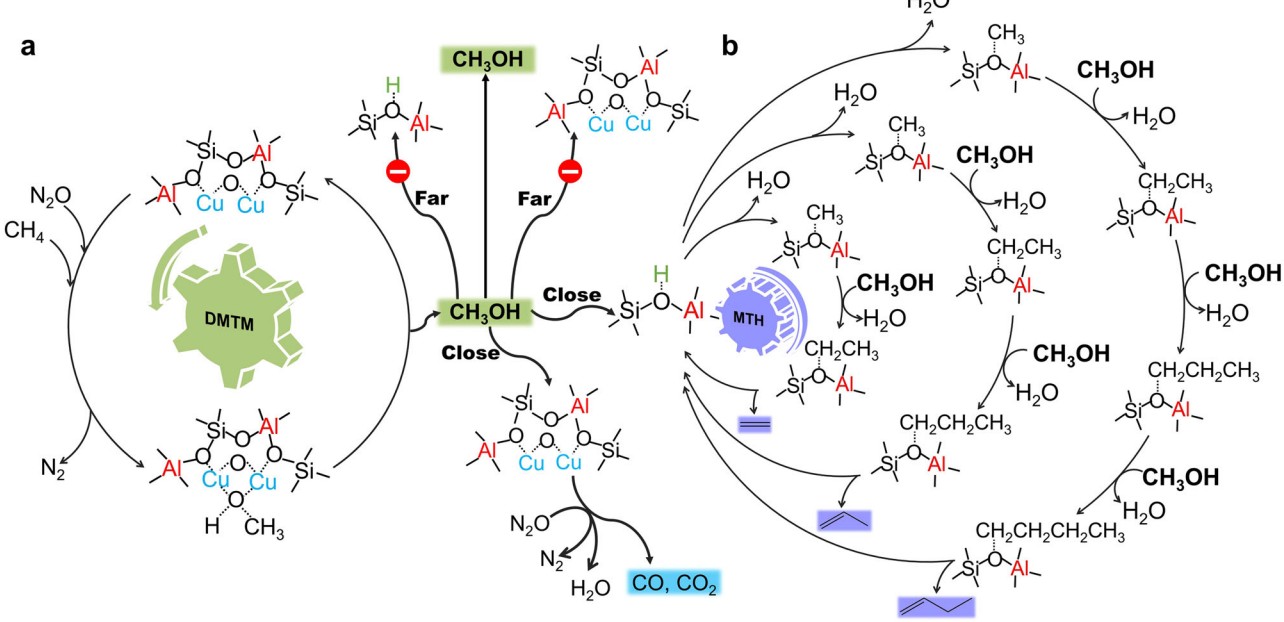

**Fig. 5 | Possible reaction pathway of direct oxidation of methane to hydrocarbons via MeOH/DME intermediates. a** Methane is directly converted to methanol (DMTM) on active Cu sites and (**b**) methanol is converted to hydrocarbons (MTH) on acid sites.

when the spatially separated Cu and Cu sites is close enough, methanol that is released from the Cu sites is possibly over-oxidized to CO and $CO_2$ on the adjacent Cu sites (Fig. 1c, d) and Supplementary Figs. 18c, d, 19c, d, 29a, b).

We have successfully revealed the effect of spatially separated Cu and BAS on the reaction performance of direct oxidation of methane by ion exchange of different concentrations of $Cu(NO_3)_2$ solution with AEI zeolite, and by physical mixture of Cu/AEI with acidic zeolite in different integration manners. It has been exposed that the low Cu density in Cu/AEI zeolite implied the closely separated Cu and BAS, while the high Cu density in Cu/AEI zeolite meant the closely separated Cu and Cu sites and thus the distantly separated Cu and BAS. Closely located Cu and BAS were beneficial for forming hydrocarbons due to the tandem conversion of methanol/ DME on the nearby acid sites. Closely located Cu and Cu sites were prone to over-oxidize methanol to $CO_2$ on the adjacent Cu sites. The steady and efficient production of methanol from methane was based on the trade-off of spatially separated Cu and BAS. In addition, the physical mixture Cu/AEI with acidic zeolite in different integration manners also exhibited that the closely separated Cu and BAS was advantageous for the generation of hydrocarbons, and the product distribution was highly dependent on the acidic properties and structure of acidic zeolites. These findings will guide to design of efficient catalysts to control side reactions and increase methanol yield in direct oxidation of methane to methanol reaction, as well as open up an avenue for direct oxidation of methane to hydrocarbons at low temperatures.

## Methods
### Catalyst preparation
***x*Cu/AEI**. Aluminosilicate AEI-type zeolite ($NH_4^+$ form, Si/Al=6.9) provided by BASF SE was used as the parent zeolite[56]. The Cu density was regulated by using the parent zeolite to ion exchange with 1, 5, 50, 500 mmol/L $Cu(NO_3)_2$ (FUJIFILM, Wako Special Grade) solutions, respectively, at 80 °C stirring for 24 h (Solid/Liquid ratio was 1 g/100 ml). The solid products were washed, dried at 100 °C overnight, and calcined at 550 °C for 5 h in air. The obtained samples were denoted as *x*Cu/AEI, where *x* means the $Cu(NO_3)_2$ concentration.

**5Cu/AEI-*t*.** To adjust the acid density, the dried 5Cu/AEI was calcined at 550, 750, and 850 °C, respectively, for 5 h in air. The obtained samples were denoted as 5Cu/AEI-*t*, where *t* means the calcination temperature. Note that, 5Cu/AEI and 5Cu/AEI-550 are the same sample.

**Acidic zeolites.** The H-type aluminosilicate AEI (H-AEI), MFI (H-MF), BEA (H-BEA) zeolites were prepared by calcination AEI (BASF SE, $NH_4^+$ form), MFI (Japan Reference Catalyst, JRC-Z5-30NH4, $NH_4^+$ form), Beta (Zeolyst International, CP814E*, $NH_4^+$ form) zeolites at 550 °C for 5 h in air.

### Characterization of catalysts
XRD pattern was collected on a Rint-Ultima III (Rigaku) using a Cu Kα X-ray source (40 kV, 40 mA).

Elemental analysis was performed on an inductively coupled plasma-atomic emission spectrometer (ICP-AES, Shimadzu ICPE-9000).

Field-emission scanning electron microscopic (FE-SEM) images of the powder samples were obtained on SU9000 (Hitachi) microscope operating at 1 kV.

High-angle annular dark field scanning transmission electron microscopy (HAADF-STEM) imaging and energy dispersive spectrometry (EDS) mapping were performed on a FEI Themis Z microscope equipped with an XFEG field electron source and double aberration corrector operated on 300 keV. The HAADF-STEM images were acquired with camera length as 115 mm while the beam convergence was 25.1 mrads. The pixle size is 37 pm and the dwell time is 2 us/pixle. The collection angle of HAADF detector was set to 48–200 mrads. STEM-EDS elemental maps were acquired with 4 us/pixel acquisition time using Velox software

Nitrogen adsorption and desorption measurements to obtain information of the micro- and meso-porosities were conducted at −196 °C on a Belsorp-mini II (MicrotracBEL).

The amount of coke in the spent samples was determined by the weight loss from 250 to 800 °C in a thermogravimetric (TG) profile, which was performed on a thermogravimetric-differential thermal analyzer (TG-DTA, RigakuThermo plus EVO II).

Solid-state $^{29}Si$ and $^{27}Al$ MAS NMR spectra were measured on a JEOL ECA-600 spectrometer at a resonance frequency of 156.4 MHz

using a 4 mm sample rotor with a spinning rate of 15.0 kHz. The $^{29}Si$ and $^{27}Al$ chemical shifts were referenced to −34.12 and −0.54 ppm, polydimethylsiloxane (PDMS) and $AlNH_4(SO_4)_2 \cdot 12H_2O$, respectively.

Temperature-programmed ammonia desorption ($NH_3$-TPD) profiles were recorded on Multitrack TPD equipment (Japan BEL). Typically, 25 mg of catalyst was pretreated at 600 °C in He (50 mL·min⁻¹) (Taiyo Nippon Sanso Co., Ltd., 99.99995%) for 1 h and then cooled to 100 °C. Prior to the adsorption of $NH_3$ (Toho Chemical Industry Co., Ltd., 28.8% $NH_3$ in He), the sample was evacuated at 100 °C for 1 h. Approximately 2500 Pa of $NH_3$ was allowed to contact with the sample at 100 °C for 10 min. Subsequently, the sample was evacuated to remove weakly adsorbed $NH_3$ at the same temperature for 30 min. Finally, the sample was cooled to 100 °C and heated from 100 to 600 °C at a ramping rate of 10 °C min⁻¹ in a He flow (50 mL·min⁻¹). A thermal conductivity detector (TCD) and BELMass (MicrotacBEL Corp.) were used to monitor desorbed $NH_3$. The amount of acid sites was determined by the fitting peak area of the profiles.

UV-vis spectra were collected in the range of 190–900 nm on a V-650DS spectrometer (JASCO). The diffuse reflectance spectra were converted into the absorption spectra using the Kubelka-Munk function.

Fourier Transform Infrared (FTIR) spectra were obtained by using a JASCO 4100 FTIR spectrometer equipped with a triglycine sulfate (TGS) detector. IR spectra of the clean disk were recorded in vacuo at 25 °C to obtain background spectra. The sample was pressed into a self-supporting disk (20 mm diameter, 30 – 60 mg) and placed in an IR cell attached to a closed-gas circulation system. The sample was pretreated by evacuation at 500 °C for 1 h, followed by measuring the hydroxyl vibration at −120 °C and then adsorption of 5 – 120 Pa NO (Taiyo Nippon Sanso Co., Ltd., 99%) or 5–1000 Pa CO (Taiyo Nippon Sanso Co., Ltd., 99.95%) at −120 °C. The IR spectra resulting from the subtraction of the spectra before adsorption from those with NO or CO adsorption are shown unless otherwise noted.

## Catalyst tests

The continuous oxidation of methane reaction was performed in a fixed-bed flow reactor. The online-reaction-analysis system was equipped with two six-port inlet valves. In a typical test, 100 mg of catalyst in a granular form (400 – 600 μm) was charged into a quartz tube (inner diameter 4 mm), which was placed in an electric tube furnace. The catalyst was pretreated at 500 °C for 1 h in the Argon flow. The reaction was conducted at 350 °C in a flowing gas mixture of $CH_4$ (Taiyo Nippon Sanso Co., Ltd., Ltd., 99.999%), $N_2O$ (Koike Precision Instruments, 99.99%), $H_2O$ (Homemade deionized water), and Ar (Toho Chemical Industry Co., Ltd., General industrial) with flow rates of 10, 10, 2, and 3 mL·min⁻¹. The outlet gas, containing products, unreacted $CH_4$ and $N_2O$, was analyzed by two on-line gas chromatographs (GC; GC-2014, Shimadzu). One of the GCs was used with a Shin carbon ST 50/80 packed column (Agilent Technologies, inner diameter 3 mm, length 6 m) and a TCD detector. Specifically, GC-TCD was used to detect $H_2$, $N_2O$, CO, $CO_2$ and $CH_4$. The other GC was equipped with a Porapak Q 80/100 packed column (Agilent Technologies, inner diameter 3 mm, length 6 m), a flame ionization detector (FID), and a methanizer. The GC-FID was used to investigate $CH_4$, and the produced methanol (MeOH), dimethyl ether (DME), alkanes, and alkenes. The yield of each carbon-containing product was calculated by considering the number of carbon atoms. The methane conversion in this study was defined as the total obtained products, and calculated as:

$$C_{CH4} = \frac{\sum(i*Ci)}{\sum(i*Ci) + CH4} \tag{1}$$

where $C_{CH4}$ is the $CH_4$ conversion, $i$ is the number of carbon atoms in product $C_i$, $\Sigma(i*C_i)$ is the total amount of carbon of all the products, and $CH4$ is the amount of $CH_4$ detected at the same time.

The $N_2O$ conversion was calculated as:

$$C_{N2O} = \frac{ni - na}{ni} \tag{2}$$

where $C_{N2O}$ is the $N_2O$ conversion, $n_i$ is the initial $N_2O$ molar weight, $n_a$ is the $N_2O$ molar weight after reaction.

Note that, $CH_4$ and $N_2O$ conversion were calculated according to different methods, thus they were not equal in value.

The product selectivity was calculated as:

$$S_{Ci} = \frac{i*Ci}{\sum(i*Ci)} \tag{3}$$

where $S_{Ci}$ is the selectivity of the product $C_i$, $\Sigma(i*C_i)$ is the total amount of carbon of all the products.

The product yield was calculated as:

$$Y_{Ci} = \frac{i*Ci}{\sum(i*Ci) + CH4} \tag{4}$$

where $Y_{Ci}$ is the yield of the product $C_i$, $\Sigma(i*C_i)$ is the total amount of carbon of all the products, and $CH4$ is the amount of $CH_4$ detected at the same time.

The product formation rate was calculated as:

$$R_{Ci} = Y_{Ci}*F_{CH4}/m_{cat} \tag{5}$$

where $R_{Ci}$ is the formation rate of product $C_i$, $F_{CH4}$ is the initial flow rate of $CH_4$, $m_{cat}$ is the mass of catalyst. The total formation rate $r_{CH4} = r_{CO} + r_{CO2} + r_{CH3OH} + 2*r_{DME} + 2*(r_{C2H4} + r_{C2H6}) + 3*(r_{C3H6} + r_{C3H8}) + 4*(r_{C4H8} + r_{C4H10}) + 5*r_{C5+}$.

## Statistics and reproducibility

We repeated the main catalysts for the catalyst test. All the experimental results were able to be reproduced within a small margin of error. No statistical method was used to predetermine the sample size. No data were excluded from the analyses. The experiments were not randomized. The investigators were not blinded to allocation during experiments and outcome assessment.

## Reporting summary

Further information on research design is available in the Nature Portfolio Reporting Summary linked to this article.

## Data availability

The source data generated in this study are provided in the Source Data file. Source data are provided with this paper.

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

## Acknowledgements
This work was supported by the JSPS KAKENHI Grant-in-Aid for Scientific Research (B) (No. 21H01714) and JSPS KAKENHI Grantin-Aid for Scientific Research (S) (No. 21H05011).

## Author contributions
P.X. carried out the experiments on the synthesis, characterization of the materials, catalytic reaction test, data processing, data visualization, manuscript writing and revision. K.N. maintained the methane reaction system. H.T. and Y.L. measured the $^{27}Al$ and $^{29}Si$ MAS NMR. L.W. and J.H. measured the HRTEM, HAADF-STEM, EDS mapping images. S.B. revised the English writing. T.Y., Y. W. and H.G. revised the manuscript. All authors provided critical feedback and helped shape the research, analysis and manuscript.

## Competing interests
The authors declare no competing interests.
