## [Peer Review File · Nature Communications]

Understanding the effect of spatially separated Cu and acid sites in zeolite catalysts on oxidation of methaneREVIEWER COMMENTS

Reviewer #1 (Remarks to the Author):

Xiao et al. report an interesting study on the tandem conversion of methane to hydrocarbons using N₂O and an oxidant and copper-containing zeolites as the catalysts. The authors achieved considerable selectivity to C₂+ compounds over several hours TOS. The results are truly novel and important, since the interest to the direct conversion of methane to valuable commodities is constantly growing. Equally, however, the manuscript should be further improved to facilitate the reading and remove inconsistencies.

1. The authors talk about the density and distribution of copper in copper-exchanged AEI without considering Al zoning and Al pairing. These features associated with zeolite synthesis, affect the location of the ion exchange positions, and in turn, the location of copper and BAS in the final samples. The authors should carefully reconsider their "statistical" assumption about Cu distribution.
2. The authors claim that they "probe the distance" between Cu and BAS, even adding this to the title. I understand, that this is ultimate goal of the study, equally, however, I did not find any evidence for the distance, directly measured in this work. I urge the authors to make clear statements, that, instead of "probing distance", they made a study on an effect of mixing of redox component (CuZEO) and acidic component (ZEO). Otherwise, the title and the text are misleading to the readership.
3. Figures are hard to read, they contain several bar graphs, which are pale and confusing. I suggest to limit the amount of information in each figure, but present it in a very clear way. I would remove the insets with re-scaled selectivity to HC.
4. The authors discuss extensively the selectivity in the reaction, setting over-oxidation against the formation of HC. To understand better the performance of the catalyst, I would like to see the data on the N₂O and methane conversion. How efficient is this tandem system?
5. The authors associate the deactivation of the catalysts with the accumulation of coke. However, under the utilized conditions (300C steam), zeolites can undergo dealumination, losing their acidity. Did authors try to regenerate their catalysts? What does ²⁷Al NMR after reaction shows?
6. The authors use ammonia TPD to measure the amount of acid sites in CuZEO. It is well-known, that copper catalyses the decomposition of ammonia, leading to incorrect TPD data. Did the authors consider this issue in their measurements? If not, TPD with MS detector might be helpful.
7. More experimental details should be added to enable reproduction of the data. For instance, how water is delivered to the system? How much? The authors report the flow, but it is not clear, under which conditions it was measured.
8. I suggest careful proofreading of the final text to remove bad expressions and typos before next submission. Unfortunately, current version contains plenty of them.

Reviewer #2 (Remarks to the Author):

This manuscript describes probing distance effect between Cu and acid sites in zeolite catalysts for oxidation of methane, where the distance between Cu and BAS was regulated by varying the

concentrations of $\text{Cu}(\text{NO}_3)_2$ solution in the Cu loading process by ion-exchange method. It was observed that the equilibrium distance between Cu and BAS was favorable for the stable production of methanol. These results are interesting, providing a guidance for designing efficient catalysts to maintain high methanol yield. Therefore, I recommend for publication of this work after minor revision in the following:

As shown in Figure 4a-4c, the desorption peak of ammonia was different, which was generally explained by different acidic strength. There is another explanation, which is due to the different diffusion rate of ammonia in the different micropore sizes. In this case, acidic strength might be very similar.

Reviewer #3 (Remarks to the Author):

The work is interesting, and is based on an impressive amount of data, including characterization that should support the hypothesis. The manuscript deserves publication, but there are some weak points concerning the experimental evidences to the working hypothesis, as detailed in the following.

The EDS elemental mapping images reported in the manuscript are too small and there is no marker in the magnification so that it is not possible to understand what one is seeing as coloured spots. It is unlikely that the green and red spots in the magnification have atomic resolution. A Si/Al = 6.9 should imply a much higher Al concentration in the catalysts than what guessed from the images reported in Figure 1 a) - d). Please improve the figure and explain better.

As a general comment, it is true that the reported characterization suggests that there is no massive aggregation of Cu in the samples, but this does not mean that small clusters of Cu oxide are not formed, since small ones would not be detected though the reported technique. The authors talk about dispersion in general, but in their proposed mechanisms seem to imply that the dispersion is at atomic scale, that is all Cu ions are counterions stabilizing the negative charge of the framework. I find this a bit misleading.

Indeed, the authors state that no aggregation of Cu is observed after high thermal treatments. But if dealumination is obtained (as indicated by the decrease of Bronsted sites observed by IR in the OH stretching region), one would wonder how Cu ions are stabilized as counterions. In Figure 22 of the supplementary material the authors comment on the intensity of the bands assigned to $\text{Cu}^{2+}(\text{NO})$ which are not affected, but don't mention the fact that the bands related to Cu^+ are strongly depleted. This would suggest a loss of accessible Cu ions.

Talking about Cu/BEA the authors state: 'However, the broad adsorption band at 800 nm assigned to the $d \rightarrow d$ transition was clearly observed, demonstrating that the Cu clusters were formed on 5Cu/BEA zeolite (Supplementary Fig. 26d).'

I don't see many differences among the spectra of Cu/BEA reported in Fig 26d and those related to the

Cu/AEI samples. Moreover, the band around 800 nm, being related to d-d transitions, is a fingerprint of Cu²⁺ ions, and cannot be used to establish the presence of Cu clusters, since it is present in all Cu-zeolites, with position and intensity depending on coordination and geometry. Also, the mention to Cu cluster is misleading: are the authors referring to metal clusters (very unlikely!), it is more likely to have small Cu oxide or hydroxide clusters/nanoparticles

The experiment with mixed catalysts is very interesting. However, when the authors grind the particles they have particles with size 200 x 600 nm. thus, the separation of Cu and Bronsted sites is in the order of hundreds of nm, which is orders of magnitude higher with respect to the separation of the same sites in a zeolite.

Dec. 22, 2023

Manuscript ID: NCOMMS-23-45343
Manuscript title: Understanding the distance effect between Cu and acid sites in zeolite catalysts for oxidation of methane
Authors: Peipei Xiao, Yong Wang, Lizhuo Wang, Hiroto Toyoda, Kengo Nakamura, Samya Bekhti, Yao Lu, Jun Huang, Hermann Gies, Toshiyuki Yokoi

Reviewer #1 (Remarks to the Author):

Xiao et al. report an interesting study on the tandem conversion of methane to hydrocarbons using N₂O and an oxidant and copper-containing zeolites as the catalysts. The authors achieved considerable selectivity to C₂+ compounds over several hours TOS. The results are truly novel and important, since the interest to the direct conversion of methane to valuable commodities is constantly growing. Equally, however, the manuscript should be further improved to facilitate the reading and remove inconsistencies.

Reply: We would thank reviewer 1 for giving the positive comments very much. Now we answer reviewer 1's comments one by one as follows:

Q1. The authors talk about the density and distribution of copper in copper-exchanged AEI without considering Al zoning and Al pairing. These features associated with zeolite synthesis, affect the location of the ion exchange positions, and in turn, the location of copper and BAS in the final samples. The authors should carefully reconsider their "statistical" assumption about Cu distribution.

A1: Thank the comments from reviewer 1. Yes, we agree with reviewer 1, and we also specifically studied the effects of Al distribution (Applied Catalysis B: Environmental 325 (2023) 122395) and Al content in the framework (ACS Catal. 2023, 13, 11057–11068) on Cu species and thus in the reaction performance of methane oxidation in our recent works. In this work, we tried to reveal the distance effects between Cu and BAS in Cu/AEI zeolites, which is quite difficult to quantify. Therefore, we built a simple model based on the cell parameters and the chemical composition, and most importantly, assumed all elements are uniformly distributed. It is just to illustrate the hypothesis of the distance effect, but in fact, the distribution of aluminum and copper is not so uniform. To quantify the distance, we introduced Cu/AEI as the copper source and H-AEI as the BAS source in the following manuscript and adjusted the distance between Cu and BAS by regulating the integration manners of the two samples. As a result, the distance effect is more visible.

Q2. The authors claim that they "probe the distance" between Cu and BAS, even adding this to the title. I understand, that this is ultimate goal of the study, equally, however, I did not find any evidence for the distance, directly measured in this work. I urge the authors to make clear statements, that, instead of "probing distance", they made a study on an effect of mixing of redox component (CuZEO) and acidic component (ZEO). Otherwise, the title and the text are misleading to the readership.

A2: Thank reviewer 1 for reading and understanding our manuscript carefully. As mentioned above, we intend to quantify the distance between Cu and BAS in Cu/AEI zeolites, however, it is quite difficult to realize until now. By changing the density of Cu and BAS, respectively, it is easy to understand that the variation of the density of Cu and BAS is essentially the change in distance between Cu and BAS.

Even though we did not provide the accurate values of distance between Cu and BAS, the variation of Cu density and strong acid amount in Cu/AEI zeolites, as well as the physical mixture of Cu/AEI zeolite and acidic AEI, MFI, and BEA zeolites in different manners, the distance between Cu and acid sites were adjusted in different levels. Based on the suggestion of reviewer 1, we changed the title from “Probing distance effect between Cu and acid sites in zeolite catalysts for oxidation of methane” to “**Understanding the** distance effect between Cu and acid sites in zeolite catalysts for oxidation of methane”.

Q3. Figures are hard to read, they contain several bar graphs, which are pale and confusing. I suggest to limit the amount of information in each figure, but present it in a very clear way. I would remove the insets with re-scaled selectivity to HC.

A3: We are very sorry that our figures do not provide you with a good reading experience. Since the product distribution, and formation rate of methanol and hydrocarbons among the x Cu/AEI samples are different. We hope to reflect as much reaction performance information as possible in the figures, so the figures may be a bit complicated at first glance. We have moved the distribution of hydrocarbons in Figs. 1, 3, 4 to the supporting information as Supplementary Figs. 33, 34, 35.

Q4. The authors discuss extensively the selectivity in the reaction, setting over-oxidation against the formation of HC. To understand better the performance of the catalyst, I would like to see the data on the N₂O and methane conversion. How efficient is this tandem system?

A4: Thank the comments from reviewer 1. The conversion of N₂O and CH₄ for all the samples has already been presented in the manuscript and the supporting information. In our reaction system, the reaction of methane to methanol (MTM) and methanol to hydrocarbons (MTH) depends on the bifunctional zeolite catalysts in the same reactor. We did not evaluate the promotional effect of MTH on MTM or methane conversion.

Q5. The authors associate the deactivation of the catalysts with the accumulation of coke. However, under the utilized conditions (300C steam), zeolites can undergo dealumination, losing their acidity. Did authors try to regenerate their catalysts? What does ²⁷Al NMR after reaction shows?

A5: Thank the comments from reviewer 1. We agree with reviewer 1 that it is possible to dealuminate during the activation and reaction process. We have compared the ²⁷Al MAS NMR spectra of fresh and spend x Cu/AEI samples (Figure 1). It can be seen that the bands for low Cu content zeolite, 1Cu/AEI and 5Cu/AEI, at 40-50 ppm and 80-60 ppm, attributed to the distorted Al in the framework, and at 40-20 ppm, assigned to penta-coordination Al on the extra framework, become broadened (Figs. 1(a and b)). While, with the concentration of the exchanged Cu solution increasing to 50 and 500

mmol/L, the degree of these bands broadening is greatly reduced, that to say that Al in 50Cu/AEI and 500Cu/AEI are relatively stable during the reaction (Figs. 1(c and d)). The reason is the stabilizing effect of cations, such as Cu^{2+} and Na^+ , on Al in the framework. (see: 1. ACS Catal. 2019, 9, 365–375. 2. ACS Catal. 2015, 5, 6078–6085. 3. Angew Chem Int Ed, 2017, 56, 3256-3260). The coke deposit amount of $x\text{Cu}/\text{AEI}$ zeolite catalysts decreased from 9.4 to 1.3 % with the Cu content increasing (Figure 2).

We have compared the NH_3 -TPD results for the fresh and spent samples of $x\text{Cu}/\text{AEI}$ (Figure 3, Table 1). The acid amounts of 1Cu/AEI and 5Cu/AEI are much lower than the fresh ones due to the carbon deposits (Figure 2) and dealuminate (Figure 1). After regeneration by calcination in air at 550 °C for 5 h, the acid amount of 1Cu/AEI and 5Cu/AEI recovered a lot but is still less than the fresh one (Table 1). In the case of 50Cu/AEI and 500Cu/AEI, due to the extremely low carbon deposits (1.3%) (Figure 2) and the stabilizing effect of Cu cations (Figure 1), the acid amounts of spent and regenerated samples are close to the fresh ones (Figs. 3 and 4).

As for the reaction performance after regeneration, we compared the fresh and regenerated 1Cu/AEI samples (Figure 5). Satisfactory stability of 1Cu/AEI was obtained in the methane conversion. The reduced acid amount of regenerated 1Cu/AEI zeolite (Figure 4) resulted in the reduced formation rate and selectivity of hydrocarbon, and thus the increased formation rate and selectivity of methanol.

Figure 1. Compare the ^{27}Al MAS NMR results of fresh and spent (a) 1Cu/AEI, (b) 5Cu/AEI, (c) 50Cu/AEI, and (d) 500Cu/AEI zeolite catalysts.

Figure 2. (a) TG analysis and (b) DTA curves of the spent x Cu/AEI zeolites.

Figure 3. Compare the NH₃-TPD patterns of fresh and spent xCu/AEI zeolite catalysts.

Figure 4. Compare the NH₃-TPD patterns of fresh and regenerated xCu/AEI zeolite catalysts by calcination at 550 °C for 5 h.

Table 1. Acid amount of the zeolites measured by NH₃-TPD.

Sample	Equipment	Acid amount (mmol/g) ^a			
		Weak	Medium	Strong	Total
1Cu/AEI	NH ₃ -TPD-A	0.51	0.49	0.55	1.55
5Cu/AEI		0.42	0.67	0.36	1.45
50Cu/AEI		0.36	0.79	0.40	1.55
500Cu/AEI		0.26	1.09	0.33	1.68
1Cu/AEI(spent) ^b	NH ₃ -TPD-A	0.25	0.27	0.44	0.96
5Cu/AEI(spent)		0.26	0.46	0.37	1.09
50Cu/AEI(spent)		0.34	0.78	0.40	1.52
500Cu/AEI(spent)		0.27	0.97	0.25	1.49
1Cu/AEI-Re ^c	NH ₃ -TPD-A	0.45	0.48	0.40	1.33
5Cu/AEI-Re		0.34	0.68	0.33	1.35
50Cu/AEI-Re		0.35	0.82	0.42	1.59
500Cu/AEI-Re		0.26	0.96	0.24	1.46

^a Determined by NH₃-TPD; fitting curves of the weak, medium, and strong acid amount were calculated at approximately 150, 300 and 400-450 °C, respectively.

^b *x*Cu/AEI(spent) represented the spent *x*Cu/AEI samples.

^c *x*Cu/AEI-Re represented the regenerated *x*Cu/AEI by calcination at 550 °C for 5 h.

Figure 5. Compare the reaction performance of (a) fresh 1Cu/AEI zeolite and (b) regenerated 1Cu/AEI zeolite by calcination at 550 °C for 5 h. Reaction conditions: 100 mg catalyst, 350 °C, CH₄/N₂O/H₂O/Ar = 10/10/2/3 ml·min⁻¹.

Q6. The authors use ammonia TPD to measure the amount of acid sites in CuZEO. It is well-known, that copper catalysis the decomposition of ammonia, leading to incorrect TPD data. Did the authors consider this issue in their measurements? If not, TPD with MS detector might be helpful.

A6: Thank the comments from reviewer 1. We used NH₃-TPD to measure the acidity of Cu-zeolite based on an extensive literature survey (1. Journal of Catalysis 161, 597–604 (1996). 2. Chemical Engineering Science 190 (2018) 60–67. 3. Microporous and Mesoporous Materials 111 (2008) 80–88, et al.) and experimental results in line with expectations (see Figure 3 in this response) (1. The amount of strong acid sites decreased with Cu content increasing due to the replacement of H from Si(OH)Al by Cu cations. 2. The amount of medium acid sites increased with Cu content because this acid site was related to Cu sites). To be more convincing, the results of NH₃-TPD-MS for xCu/AEI samples are provided (Figure 6). The mass-to-charge ratio (m/z) of 16 was adopted to represent NH₃, since the signal of $m/z = 17$ may be from NH₃ or H₂O, while $m/z = 16$ came substantially from NH₃ (Catal. Sci. Technol., 2022, 12, 4169–4180). The signal of $m/z = 28$, representing N₂, is very weak and almost shows no difference with Cu content increasing (Figure 6). Besides, we calculated and compared the acid amount of xCu/AEI using TPD and MS signal, the acid amounts were close (Table 2). Therefore, we concluded that NH₃ was not decomposed in the process of NH₃-TPD by Cu/AEI zeolites.

Figure 6. NH₃-TPD-MS profiles of (a) 1Cu/AEI, (b) 5Cu/AEI, (c) 50Cu/AEI, (d) 500Cu/AEI zeolite catalysts.

Table 2. Acid amount of the zeolites measured by NH₃-TPD-MS and NH₃-TPD.

Sample	Equipment	Acid amount (mmol/g) ^a			
		Weak	Medium	Strong	Total
1Cu/AEI	NH ₃ -TPD-A-MS	0.54	0.52	0.58	1.64
5Cu/AEI		0.37	0.61	0.33	1.31
50Cu/AEI		0.34	0.76	0.37	1.47
500Cu/AEI		0.25	1.05	0.25	1.55
1Cu/AEI	NH ₃ -TPD-A	0.51	0.49	0.55	1.55
5Cu/AEI		0.42	0.67	0.36	1.45
50Cu/AEI		0.36	0.79	0.40	1.55
500Cu/AEI		0.26	1.09	0.33	1.68

^a Determined by NH₃-TPD; fitting curves of the weak, medium, and strong acid amount were calculated at approximately 150, 300 and 400-450 °C, respectively.

Q7. More experimental details should be added to enable reproduction of the data. For instance, how water is delivered to the system? How much? The authors report the flow, but it is not clear, under which conditions it was measured.

A7: Thank the comments from reviewer 1. All the details about the reaction conditions have been presented in the manuscript and supporting information. For instance, we have listed the reactant flow rate in Figs. (1-4) as $\text{CH}_4/\text{N}_2\text{O}/\text{H}_2\text{O}/\text{Ar}=10/10/2/3 \text{ ml}\cdot\text{min}^{-1}$, and the steaming flow rate is $2 \text{ ml}\cdot\text{min}^{-1}$. About the variation of flow rate, we have presented the details in Supplementary Figure 18.

Q8. I suggest careful proofreading of the final text to remove bad expressions and typos before next submission. Unfortunately, current version contains plenty of them.

A8: Thank the comments from reviewer 1. English specialist has checked our manuscript and we have made the following changes:

1. “Here, we probed the distance effect between Cu and BAS in Cu/AEI zeolite catalysis on the reaction performance of direct oxidation of methane.”

was revised to

“Here, we **explored** the distance effect between Cu and BAS in Cu/AEI zeolite **catalysts** on the reaction performance of direct oxidation of methane.”

2. “This work would provide guidance for designing efficient catalysts to prevent methanol from reacting and thus maintain high methanol yield, as well as open up a new avenue for direct oxidation of methane to hydrocarbons (DMTH) via methanol intermediate at low temperatures.”

was revised to

“This work would **guide to design of** efficient catalysts to prevent methanol from **further** reacting and thus maintain high methanol yield, as well as open up a new avenue for direct oxidation of methane to hydrocarbons (DMTH) via methanol intermediate at low temperatures.”

3. “In addition, Deng and coworkers adopted the ^1H - ^{67}Zn double-resonance solid-state NMR technique to detect the spatial proximity/interaction between Zn^{2+} and H^+ from BAS.³⁴ However, it is still difficult to specifically and clearly quantify the distance between metal ions and BAS. ”

a) was revised to

“In addition, Deng and coworkers adopted the ^1H - ^{67}Zn double-resonance solid-state NMR technique to detect the spatial proximity/interaction between Zn^{2+} and H^+ from BAS **in Ångstrom scale**.³⁵ However, it is still difficult to **specifically quantify** the distance between metal ions and BAS **in a large-scale range**. ”

4. “In fact, in order to improve methanol selectivity, lower temperatures (200-300 °C) were regularly used to prevent MTH reaction and the over-oxidation of methanol to CO and CO_2 .^{13, 23, 24, 25}”

was revised to

“**In order to** improve methanol selectivity, lower temperatures (200-300 °C) were regularly used to prevent MTH reaction and the over-oxidation of methanol to CO and CO₂.^{13,14, 24, 25, 26}”

5. “These results would provide guidance for reasonable design catalysts and reaction conditions according to the target products.”

was revised to

“These results would **guide** reasonable design catalysts and reaction conditions according to the target products.”

6. “This is due to the fact that, MeOH is first formed on the Cu sites of Cu/AEI zeolite in the front bed, and then, the produced MeOH goes through the acid sites of Cu/AEI in the back bed and is converted to hydrocarbons.³³”

Was revised to

“**Because** MeOH is first formed on the Cu sites of Cu/AEI zeolite in the front bed, and then, the produced MeOH goes through the acid sites of Cu/AEI in the back bed and is converted to hydrocarbons.³⁰”

7. “Based on the cell parameters and the chemical composition, and most importantly, assuming that all elements are uniformly distributed, a unit cell (a=13.7 Å, b=12.6 Å, c=18.5 Å) of 1Cu/AEI and 500Cu/AEI contained 0.24 and 1.44 Cu atoms (Supplementary Table 1), respectively.”

was revised to

“Based on the cell parameters and the chemical composition, and most importantly, assuming that all elements are uniformly distributed **and ignoring Al zoning**, a unit cell (a=13.7 Å, b=12.6 Å, c=18.5 Å) of 1Cu/AEI and 500Cu/AEI contained 0.24 and 1.44 Cu atoms (Supplementary Table 1), respectively. ”

8. “Namely, 4 unit cells shared *ca.* 1 Cu atom in 1Cu/AEI and *ca.* 6 Cu atoms in 500Cu/AEI.”

was revised to

“Namely, 4 unit cells shared *ca.* 1 Cu atom in 1Cu/AEI and *ca.* 6 Cu atoms in 500Cu/AEI. **Assume that a single Cu corresponds to a single Al, and Cu pairs correspond to Al pairs.** ”

9. “The possibility of MeOH produced on Cu sites being transported to acid sites was reduced. Therefore, the maximum selectivity of hydrocarbons declined from 95 % for 1Cu/AEI to 0.3 % for 500Cu/AEI. Furthermore, the increased Cu density resulted in the declined distance between Cu and Cu sites, and thus the methanol intermediate was over-oxidized to CO and CO₂.²²”

was revised to

“The possibility of MeOH produced on Cu sites being transported to acid sites was reduced **in space**. Therefore, the maximum selectivity of hydrocarbons declined from 95 % for 1Cu/AEI to 0.3 % for 500Cu/AEI. Furthermore, the increased Cu density resulted in the **closer** distance

between Cu and Cu sites, which can be visible in the representative and comparison HAADF-STEM images of 1Cu/AEI and 500Cu/AEI (Supplementary Figs. 6(i and j)), and thus the methanol intermediate was over-oxidized to CO and CO₂.²³”

10. “Aggregated Cu clusters were not observed by increasing the calcination temperature from 550 to 850 °C (Supplementary Figs. 17-22).”

was revised to

“Aggregated Cu clusters were not observed after calcination in elevated temperatures from 550 to 850 °C (Supplementary Figs. 20-26).”

Reviewer #2 (Remarks to the Author):

This manuscript describes probing distance effect between Cu and acid sites in zeolite catalysts for oxidation of methane, where the distance between Cu and BAS was regulated by varying the concentrations of Cu(NO₃)₂ solution in the Cu loading process by ion-exchange method. It was observed that the equilibrium distance between Cu and BAS was favorable for the stable production of methanol. These results are interesting, providing a guidance for designing efficient catalysts to maintain high methanol yield. Therefore, I recommend for publication of this work after minor revision in the following:

Reply: We would thank reviewer 2 for giving the positive comments very much. Now we answer reviewer 2's comments one by one as follows:

Q1: As shown in Figure 4a-4c, the desorption peak of ammonia was different, which was generally explained by different acidic strength. There is another explanation, which is due to the different diffusion rate of ammonia in the different micropore sizes. In this case, acidic strength might be very similar.

A1: Thank the comments from reviewer 2. We agree with reviewer 2 that the pore size and dimensions of zeolites influenced the acidic strength due to the different diffusion rates. It is easier to desorb NH₃ from large pore zeolite, which is reflected in the weak acid strength. Based on the gradually decreased desorption temperature of strong acid sites from around 475 (AEI) to 400 (MFI) and 325 °C (BEA), we think it is no problem to compare the acid strength and conclude that the acid strength, especially the strong acid sites, decreased with the pore size of zeolite increasing. For better comparison, we have changed the peak fitting of MFI in Figure 7b. In addition, we have changed the sentence "The remaining MeOH and DME were the apparent indication of the insufficient strength and content of acid sites, which was confirmed by the NH₃-TPD results (Figs. 4(b, c) and Supplementary Table 2)."

to

"The remaining MeOH and DME were the apparent indication of the insufficient **acid content, which was confirmed by the NH₃-TPD results (Figs. 4(b, c) and Supplementary Table 2), and an indication of inaccessible acid sites.**"

Figure 7. Deconvolution of NH_3 -TPD curves of (a) H-AEI, (b) H-MFI, (c) H-BEA zeolites.

Reviewer #3 (Remarks to the Author):

The work is interesting, and is based on an impressive amount of data, including characterization that should support the hypothesis. The manuscript deserves publication, but there are some weak points concerning the experimental evidences to the working hypothesis, as detailed in the following.

Reply: We would thank reviewer 3 for giving the positive comments very much. Now we answer reviewer 3's comments one by one as follows:

Q1: The EDS elemental mapping images reported in the manuscript are too small and there is no marker in the magnification so that it is not possible to understand what one is seeing as colored spots. It is unlikely that the green and red spot in the magnification have atomic resolution. A Si/Al = 6.9 should imply a much higher Al concentration in the catalysts than what guessed from the images reported in Figure 1 a) - d). Please improve the figure and explain better.

A1: Thank the comments from reviewer 3. We intended to reveal the different distances between Cu and Al by HAADF-STEM and EDS mapping images, however, it is difficult. The red and green spots in the magnified images represented the Al and Cu atoms, respectively. The density of elements observed by EDS mapping images highly depends on the test duration and calculation method. It is not easy to compare different samples. Thus, we removed the magnified images in Figs. 1(a-d) in the manuscript and only kept the EDS images. In addition, we added Figure 8 to reveal the distances between Cu and Cu in 1Cu/AEI and 500Cu/AEI zeolite catalysts as Supplementary Figs. 6(i and f). The statement that the higher Cu content means the closer distance between Cu and Cu has been verified by the higher proportion (50%) at 1.0~2.0Å for 500Cu/AEI zeolite.

Figure 8. Representative HAADF-STEM images of (a) 1Cu/AEI and (b) 500Cu/AEI zeolite catalysts to partially reveal the distance between Cu and Cu distance.

Q2:As a general comment, it is true that the reported characterization suggests that there is no massive aggregation of Cu in the samples, but this does not mean that small clusters of Cu oxide are not formed, since small ones would not be detected though the reported technique. The authors talk about dispersion in general, but in their proposed mechanisms seem to imply that the dispersion is at atomic scale, that is all Cu ions are counterions stabilizing the negative charge of the framework. I find this a bit misleading.

A2: Thank the comments from reviewer 3. We completely agree with reviewer 3 in theory. The isolated, binuclear, and trinuclear Cu species can be identified by NO adsorption FTIR, and CuO nanoparticles can be detected by XRD and TEM. However, the larger polynuclear Cu complexes, such as tetra- and pentamer copper clusters, can not be detected via experiments. Their existence and activity are based on DFT calculation (see: *Nanoscale* **2017**, 9, 1144). Because we neither observed bands at 1995 cm⁻¹ in NO adsorption FTIR spectra, belonging to the trinuclear Cu species, nor detected peaks at 35.6 and 38.8° in XRD patterns, belonging to CuO nanoparticles. We thought the Cu species mainly focused on isolated and binuclear Cu species. We agree that Cu exists as the atomic scale and Cu cations can stabilize the Al in the framework, which has been mentioned above and reported in the literature (see: 1. ACS Catal. 2019, 9, 365–375. 2. ACS Catal. 2015, 5, 6078–6085. 3. Angew Chem Int Ed, 2017, 56, 3256-3260).

Q3:Indeed, the authors state that no aggregation of Cu is observed after high thermal treatments. But if dealumination is obtained (as indicated by the decrease of Bronsted sites observed by IR in the OH stretching region), one would wonder how Cu ions are stabilized as counterions. In Figure 22 of the supplementary material the authors comment on the intensity of the bands assigned to Cu²⁺(NO) which are not affected, but don't mention the fact that the bands related to Cu⁺ are strongly depleted. This would suggest a loss of accessible Cu ions.

A3: Thank the comments from reviewer 3. Only a part of H (or NH₄⁺) cations have been replaced by Cu cations. The dealumination in (Si-OH-Al) is easier to occur (see Figs. 2(f, g)). As we mentioned above (Supplementary Figure 16), the Cu²⁺ on the extra framework was helpful to stable Al in the framework (see: 1. ACS Catal. 2019, 9, 365–375. 2. ACS Catal. 2015, 5, 6078–6085. 3. Angew Chem Int Ed,2017,56, 3256-3260).

We used NO adsorption FTIR spectra to identify the Cu species based on the literature “*J. Phys. Chem. C* **2021**, 125, 12094–12106”. In this literature, only the bands at 1880-1995 cm⁻¹ are used to check the nuclear of Cu species. It is not clear about the nuclear assignment of Cu⁺ species at 1850-1650 cm⁻¹, which is formed by “auto reduction”.

As for the severely decreased intensity of Cu⁺ species bands for 5Cu/AEI-750 and 5Cu/AEI-850 in comparison with 5Cu/AEI-550, the possible reason was the Cu²⁺ species, which was easily reduced to Cu⁺, was transferred to Cu²⁺, which was not easily reduced to Cu⁺. Because the new band at 1917 cm⁻¹, assigned to the dicopper species, was observed on 5Cu/AEI-750 and 5Cu/AEI-850.

Q4:Talking about Cu/BEA the authors state: 'However, the broad adsorption band at 800 nm assigned

to the $d \rightarrow d$ transition was clearly observed, demonstrating that the Cu clusters were formed on 5Cu/BEA zeolite (Supplementary Fig. 26d). I don't see many differences among the spectra of Cu/BEA reported in Fig 26d and those related to the Cu/AEI samples. Moreover, the band around 800 nm, being related to d-d transitions, is a fingerprint of Cu^{2+} ions, and cannot be used to establish the presence of Cu clusters, since it is present in all Cu-zeolites, with position and intensity depending on coordination and geometry. Also, the mention to Cu cluster is misleading: are the authors referring to metal clusters (very unlikely!), it is more likely to have small Cu oxide or hydroxide clusters/nanoparticles

A4: Thank the comments from reviewer 3. The difference between 5Cu/AEI and 5Cu/BEA at 650-850 nm in UV-vis spectra (Supplementary Figure 30d) is obvious. For $x\text{Cu}/\text{AEI}$, the intensity of peak at 650-850 nm only slightly increased with Cu content growing from 0.47 to 2.92 wt.%. 5Cu/AEI (0.63 wt.%) and 5Cu/BEA (0.73 wt.%) have close Cu content, however, the intensity of 650-850 nm band for 5Cu/BEA was stronger than that of 5Cu/AEI. Considering no CuO nanoparticles are observed at 35.6 and 38.8° in XRD patterns, the band at 650-850 nm in UV-vis spectra with strong intensity was thought as Cu clusters.

For further confirmation, we have added the HADDF-STEM and EDS mapping images of 5Cu/BEA in Figure 9. As we can see, all the Cu species were uniformly dispersed and no severely aggregated Cu species were observed. In addition, the NO adsorption FTIR spectra of 5Cu/AEI and 5Cu/BEA zeolite at -120°C were compared in Figure 10. Compared with 5Cu/AEI, the bands belonging to $\text{Cu}^{2+}(\text{NO})$ of 5Cu/BEA shifted to high wavenumbers 1959, 1948, and 1936 cm^{-1} , which was possibly a signal of the Cu clusters according to the literature (J. Phys. Chem. C 2021, 125, 12094–12106). Figs. 9 and 10 were added as Supplementary Figs. 30 (e and f), respectively.

Figure 9. HADDF-STEM and EDS mapping images of 5Cu/BEA zeolite.

Figure 10. Compare the NO (5-120 Pa) adsorption FTIR spectra of (a) 5Cu/AEI and (b) 5Cu/BEA zeolite at -120 °C after evacuated at 500 °C for 1 h.

Q5: The experiment with mixed catalysts is very interesting. However, when the authors grind the particles they have particles with size 200 x 600 nm. thus, the separation of Cu and Bronsted sites is in the order of hundreds of nm, which is orders of magnitude higher with respect to the separation of the same sites in a zeolite.

A5: Thank the comments from reviewer 3. Yes, we agree with reviewer 3, the adjusting between Cu and BAS still has a lot of work to do. In the next step, we will exchange Cu with different structure zeolites, and consider the distance effect on the nanometer or angstrom level between Cu and BAS. In that case, we need to carefully consider the effects of Cu species, maybe the isolated Cu is better. Here, we can simply compare the Cu species, NH₃-TPD, and the reaction results of 1Cu/AEI as the close distance between Cu and BAS due to small pore size, and 1Cu/BEA as the farther distance between Cu and BAS due to the large pore size in Figure 11. Even though the Cu/Al ratio was low to 0.03 for 1Cu/BEA zeolite, the Cu cluster was formed (Figures 11(a and c)). The acid amount of 1Cu/BEA was lower than 1Cu/AEI (Figure 11d). In the case of reaction performance, the selectivity and formation rate of hydrocarbons for 1Cu/AEI reached the highest at TOS=1.16 h, and the selectivity of hydrocarbons was still up to 80% at TOS =4.16 h (Figure 11e). However, no hydrocarbons were observed for 1Cu/BEA zeolite (Figure 11f). The different acid amount was one reason. Another possible and important reason should be the farther distance between Cu and BAS.

Figure 11. (a) Compare UV-vis spectra of the 1Cu/AEI and 1Cu/BEA zeolites. FTIR spectra of adsorbed NO (5-120 Pa) at -120 °C of (b) 1Cu/AEI and (c) 1Cu/BEA zeolites. (d) Compare NH₃-TPD curves of 1Cu/AEI and 1Cu/BEA zeolites. Compare the reaction performance of (e) 1Cu/AEI and (f) 1Cu/BEA in methane oxidation reaction. Reaction conditions: 100 mg catalyst, 350 °C, CH₄/N₂O/H₂O/Ar = 10/10/2/3 ml·min⁻¹.

In addition, the following changes were made:

1. “5Cu/Beta zeolite was prepared through the same procedure with 5Cu/AEI just using Beta (Zeolyst, CP814E*, in the NH_4^+ form) as the parent zeolite.”

was revised to

“1Cu/BEA and 5Cu/BEA zeolite catalysts were prepared by the same procedure with 1Cu/AEI and 5Cu/AEI just using Beta (Zeolyst, CP814E*, in the NH_4^+ form) as the parent zeolite.”

2. “After reaction several hours, the coke deposited on or close to the acid sites resulted in the inactivation of acid sites. It is noteworthy that the activity of Cu sites was still maintained, thus the productivity of MeOH was able to recover to some extent.”

was revised to

“After reaction several hours, the coke deposited on the acid sites and resulted in the inactivation of acid sites, which can be confirmed by comparison of the results of NH_3 -TPD and TG-DTA for the fresh and spent zeolites (Supplementary Figs. 14, 15). In addition, another reason that caused the gradually declined hydrocarbon formation rate was dealumination in the process of activation and reaction, which has been approved by comparison of the ^{27}Al MAS NMR curves of the fresh and spent zeolites (Supplementary Fig. 16).” It is noteworthy that the activity of Cu sites was still maintained, thus the productivity of MeOH was able to recover (Supplementary Figs. 14(c and d)).

3. “50Cu/AEI and 500Cu/AEI showed obviously different CO_2 selectivity compared to 1Cu/AEI and 5Cu/AEI, signifying that the excessive Cu amount led to over-oxidation.²²”

was revised to

“50Cu/AEI and 500Cu/AEI showed different CO_2 selectivity compared to 1Cu/AEI and 5Cu/AEI, signifying that the excessive Cu amount led to over-oxidation.²³ Moreover, the Cu cations on the extra framework can stabilize Al in the framework of zeolites,³⁸ therefore, the NH_3 -TPD patterns and ^{27}Al MAS NMR spectra of fresh and spent 50Cu/AEI and 500Cu/AEI samples only showed a tiny difference (Supplementary Figs. 14, 16c, 16d).”

4. “One should note that the methanol formation rate of 5Cu/AEI-850 amplified and stabled at *ca.* $33 \mu\text{mol}\cdot\text{g}^{-1}\cdot\text{min}^{-1}$ ($1980 \mu\text{mol}\cdot\text{g}^{-1}\cdot\text{h}^{-1}$) due to the low selectivity of total hydrocarbons.”

was revised to

“One should note that the methanol formation rate of 5Cu/AEI-850 amplified and stabled at *ca.* $33 \mu\text{mol}\cdot\text{g}^{-1}\cdot\text{min}^{-1}$ ($1980 \mu\text{mol}\cdot\text{g}^{-1}\cdot\text{h}^{-1}$) due to the low selectivity of total hydrocarbons, which was the source of carbon deposit.”

5. “When the sample was calcined at a low temperature (550°C), most of Al atoms in the framework

was reserved.”

was revised to

“It is worth pointing out that cations such as Cu^{2+} and Na^+ on the extra framework of zeolite can stabilize Al in the framework.³⁸ Thus, dealumination via high-temperature calcination mainly occurred on Al of $\text{Si}(\text{OH})\text{Al}$ (Figs. 2(f, g)). When the sample was calcined at a low temperature (550 °C), most of Al atoms in the framework were reserved.”

6. “Simultaneously, because the Cu content and Cu states offered negligible changes (Supplementary Figs. 24,25), few MeOH and DME were over-oxidized to CO_2 (Fig. 2c).”

was revised to

“Simultaneously, because the Cu content and Cu states offered negligible changes (Supplementary Figs. 24,25), few MeOH and DME were over-oxidized to CO_2 (Fig. 2c). The carbon deposit amount of 5Cu/AEI- t decayed from 8.0 to 5.2 % with t increasing from 550 to 850 °C (Supplementary Fig. 28).”

7. “5Cu/Beta zeolite was prepared using the same method as 5Cu/AEI. Analogous to 5Cu/AEI, the XRD pattern and UV-vis spectrum of 5Cu/Beta indicated that aggregated CuO nanoparticles were not observed (Supplementary Figs. 26(c, d)). However, the broad adsorption band at 800 nm assigned to the $d \rightarrow d$ transition was clearly observed,⁴⁰ demonstrating that the Cu clusters were formed on 5Cu/BEA zeolite (Supplementary Fig. 26d).”

was revised to

“5Cu/BEA zeolite was prepared using the same method as 5Cu/AEI. Analogous to 5Cu/AEI, the XRD pattern, UV-vis spectrum, HAADF-STEM images and EDS mapping images of 5Cu/BEA indicated that aggregated CuO nanoparticles were not observed (Supplementary Figs. 30(c, d, e)). The broad adsorption band at 650-850 nm of 5Cu/BEA was assigned to the $d \rightarrow d$ transition.⁴² Combine with the NO adsorption FTIR spectra of 5Cu/BEA, the $\text{Cu}^{2+}(\text{NO})$ bands at higher wavenumbers of 1959, 1948, and 1936 cm^{-1} possibly indicated the Cu clusters in 5Cu/BEA zeolite (Supplementary Fig. 30f).”

8. “The Cu clusters, less acid amount, weaker acid strength, and the farther distance between Cu and BAS on 5Cu/Beta zeolite resulted in *ca.* 75 % CO_2 selectivity and no hydrocarbons, while the main product of 5Cu/AEI was MeOH (48 %) ((Supplementary Figs. 26(f, g)).”

was revised to

“The Cu clusters, less acid amount, and the farther distance between Cu and BAS on 5Cu/BEA zeolite resulted in *ca.* 75 % CO_2 selectivity and no hydrocarbons, while the main product of 5Cu/AEI was MeOH (48 %) (Supplementary Figs. 30(h, i)). Additionally, the compared characterization and reaction results of 1Cu/AEI and 1Cu/BEA zeolites further confirmed the statement (Supplementary Fig. 31).”

9. The changes of references, number of figures, and changes in the supplementary materials have not been described in detail here.

REVIEWER COMMENTS

Reviewer #1 (Remarks to the Author):

The authors made a substantial revision of the present manuscript, addressing most of my comments, which is very much appreciated. The only two of them require additional attention.

1. I still believe the the title (Understanding the distance effect between Cu and acid sites in zeolite catalysts for oxidation of methane) and the conclusion section do not fully reflect the actual novelty and impact of this study. The authors insist the their work studies the "distance" between the acid site and copper sine in bifunctional catalyst. I argue that this is about studying the effect of preparation - like nicely depicted in Figure 3: dual bed vs. mixture vs. grinding. I would call it varying "spatial separation" rather than "distance". Importantly, the authors themselves admit that measuring the actual distance and distribution between BAS and Cu is extremely complex if not impossible. In light of that I urge the authors to adjust the title, abstract, conclusions and associated discussion to not to mislead the potential readers.

2. I am not fully convinced by the results of NH₃-TPD-MS. I see that ammonia does not undergo decomposition to nitrogen, this is true. But in this sort of experiments with CuZEO ammonia frequently produces NO and NO₂, not N₂. Could the authors show the data for m/z = 30 and 46?

Reviewer #2 (Remarks to the Author):

After modifications, this work might be accepted for the publication.

Reviewer #3 (Remarks to the Author):

The clarity and consistency of the manuscript has increased. Minor changes are still requested before publication

Following the comment of Reviewer 1, I think the title is still misleading. Also, the mention to the fact that a statistical distribution of Al and Cu in the samples, without zoning and pairing should be clearly made in the text while introducing the approach used.

Pages 9/10: 'Assume that a single Cu corresponds to a single Al, and Cu pairs correspond to Al pairs.' The sentence is badly written, and also incorrect. Al pairs should stabilize CuII sites, while 'isolated' Al should stabilize CuOH sites

End of page 7: 'The xCu/AEI zeolite catalysts with varied Cu content and acidity ... were used to perform the intimacy effect between Cu and BAS on the reaction performance of DMTM' please revise sentence

The HAADF-STEM images in Supplementary Figs. 6(i and j)), are impressive, but it is not clear what the

authors are measuring in the corresponding graphs

Page 8: 'was dealumination in the process of activation and reaction, which has been approved by comparison of the ^{27}Al MAS NMR curves' please revise sentence

Page 9: 'Moreover, the Cu cations on the extra framework can stabilize Al in the framework of zeolites, therefore, the NH_3 -TPD patterns and ^{27}Al MAS NMR 153 spectra of fresh and spent 50Cu/AEI and 500Cu/AEI samples only showed a tiny difference (Supplementary Figs. 14, 16c, 16d)' This sentence was added in answer to the reviewers comment, and seems to refer to the stability of the catalysts. This should be better stated since it is not so clear as writtten

Page 12: 'It is worth pointing out that cations such as Cu^{2+} and Na^+ on the extra framework of zeolite can stabilize Al in the framework.³⁸ Thus, dealumination via high- temperature calcination mainly occurred on Al of $\text{Si}(\text{OH})\text{Al}$ (Figs. 2(f, g)).' This is an assumption, not an evidence

Response to Reviewers' comments

Reviewer #1 (Remarks to the Author):

The authors made a substantial revision of the present manuscript, addressing most of my comments, which is very much appreciated. The only two of them require additional attention.

1. I still believe the title (Understanding the distance effect between Cu and acid sites in zeolite catalysts for oxidation of methane) and the conclusion section do not fully reflect the actual novelty and impact of this study. The authors insist their work studies the "distance" between the acid site and copper sine in bifunctional catalyst. I argue that this is about studying the effect of preparation - like nicely depicted in Figure 3: dual bed vs. mixture vs. grinding. I would call it varying "spatial separation" rather than "distance". Importantly, the authors themselves admit that measuring the actual distance and distribution between BAS and Cu is extremely complex if not impossible. In light of that I urge the authors to adjust the title, abstract, conclusions and associated discussion to not to mislead the potential readers.

Reply 1: We would like to give our sincere thanks to reviewer 1 for the positive evaluation of our work and also for the quite valuable comments. Indeed, we used the "distance" in the MS but it is difficult to provide the accurate distance value. We have changed the title from "Understanding the distance effect between Cu and acid sites in zeolite catalysts for oxidation of methane" to "Understanding **the effect of spatially separated** Cu and acid sites in zeolite catalysts **on** oxidation of methane". The corresponding description in the manuscript has been changed and marked in red.

2. I am not fully convinced by the results of NH₃-TPD-MS. I see that ammonia does not undergo decomposition to nitrogen, this is true. But in this sort of experiments with CuZEO ammonia frequently produces NO and NO₂, not N₂. Could the authors show the data for m/z = 30 and 46?

Reply 2: Thank reviewer 1 for checking our response carefully. When we measured NH₃-TPD-MS, the gas only contained NH₃ and He. If NH₃ was decomposed, the possible products were N₂ and H₂. Reviewer 1 thought that "CuZEO ammonia frequently produces NO and NO₂". If so the source of oxygen merely came from zeolite and the amount of NO or NO₂ would be quite limited without the additional oxygen source. We remeasured 500Cu/AEI as an example due to the highest Cu content with the additional signals of m/z = 30 and 46, which are derived from NO and NO₂, respectively. The result is shown in following. The signals of m/z = 28, 30 and 46 are too low to influence the result of NH₃-TPD. Therefore, it can be inferred that NH₃ does not undergo decomposition to N₂ or oxidation to NO or NO₂ on Cu/AEI zeolite during the NH₃-TPD measurement.

Reviewer #2 (Remarks to the Author):

After modifications, this work might be accepted for the publication.

We would like to give our sincere thanks to reviewer 2 for the positive evaluation of our work.

Reviewer #3 (Remarks to the Author):

The clarity and consistency of the manuscript has increased. Minor changes are still requested before publication

Following the comment of Reviewer 1, I think the title is still misleading. Also, the mention to the fact that a statistical distribution of Al and Cu in the samples, without zoning and pairing should be clearly made in the text while introducing the approach used.

Reply 1: We would like to give our sincere thanks to reviewer 3 for the positive evaluation of our work and also for the quite valuable comments. We have changed the title from “Understanding the distance effect between Cu and acid sites in zeolite catalysts for oxidation of methane” to “Understanding **the effect of spatially separated** Cu and acid sites in zeolite catalysts **on** oxidation of methane”. The corresponding description in the manuscript has been changed and marked in red. In addition, in Page 9, we have mentioned “assuming that all elements are uniformly distributed and ignoring Al zoning”

Pages9/10: 'Assume that a single Cu corresponds to a single Al, and Cu pairs correspond to Al pairs.' The sentence is badly written, and also uncorrect. Al pairs should stabilize CuII sites, while 'isolated' Al should stabilize CuOH sites

Reply 2: Thank the comments from reviewer 3. We agree with reviewer 3 that our statement “Assume that a single Cu corresponds to a single Al, and Cu pairs correspond to Al pairs.” is not accurate enough. Since it was reported that Al pairs can stabilize CuII (Nature Communications (2023) 14:7705), Cu pairs (J. Am. Chem. Soc. 2019, 141, 11641–11650), tri-copper (Nature Communications (2015) 6,7546), tetra- or pentameric copper species (Nanoscale, 2017, **9**, 1144-1153). Thus, the possible combinations between Al and Cu species are quite complex. For simplicity, we only mentioned two possible situations.

End of page 7: 'The xCu/AEI zeolite catalysts with varied Cu content and acidity ... were used to perform the intimacy effect between Cu and BAS on the reaction performance of DMTM' please revise sentence

Reply 3: Thank the careful check from reviewer 3. The following sentence has been revised.

Before: “The xCu/AEI zeolite catalysts with varied Cu content and acidity (Supplementary Figs. 8-11 and Supplementary Tables 1-3) were used to perform the intimacy effect between Cu and BAS on the reaction performance of DMTM (Supplementary Figs. 12,13).”

After: “The xCu/AEI zeolite catalysts with **varying** Cu content (Supplementary Figs. 8-11 and Supplementary Tables 1-3) were used to perform the intimacy effect between Cu and BAS on the reaction performance of DMTM (Supplementary Figs. 12,13).”

The HAADF-STEM images in Supplementary Figs. 6(i and j)), are impressive, but it is not clear what the authors are measuring in the corresponding graphs

Reply 4: Thank the comment from reviewer 3, in the icon of Supplementary Figs. 6, we have mentioned “partially reveal the distance between Cu and Cu.” To be more eye-catching, we have marked in Supplementary Figs. 6(i and j)).

Page 8: 'was dealumination in the process of activation and reaction, which has been approved by comparison of the ²⁷Al MAS NMR curves' please revise sentence

Reply 5: Thank the careful check from reviewer 3. The following sentence has been revised.

Before: “In addition, another reason that caused the gradually declined hydrocarbon formation rate was dealumination in the process of activation and reaction, which has been approved by comparison of the ²⁷Al MAS NMR curves of the fresh and spent zeolites (Supplementary Fig. 16).”

has been revised to

After: “In addition, another reason that caused the gradually declined hydrocarbon formation rate was dealumination in the process of activation and reaction, which has been **proven** by comparison of the ²⁷Al MAS NMR curves of the fresh and spent zeolites (Supplementary Fig. 16).”

Page 9: 'Moreover, the Cu cations on the extra framework can stabilize Al in the framework of zeolites, therefore, the NH₃-TPD patterns and ²⁷Al MAS NMR spectra of fresh and spent 50Cu/AEI and 500Cu/AEI samples only showed a tiny difference (Supplementary Figs. 14, 16c, 16d)' This sentence was added in answer to the review comment, and seems to refer to the stability of the catalysts. This should be better stated since it is not so clear as written

Reply 6: Thank the careful check from reviewer 3. We have deleted the sentence “Moreover, the Cu cations on the extra framework can stabilize Al in the framework of zeolites,³⁸ therefore, the NH₃-TPD patterns and ²⁷Al MAS NMR spectra of fresh and spent 50Cu/AEI and 500Cu/AEI samples only showed a tiny difference (Supplementary Figs. 14, 16c, 16d).” in Page 9 since it is not suitable here.

Page 12: 'It is worth pointing out that cations such as Cu²⁺ and Na⁺ on the extra framework of zeolite can stabilize Al in the framework.³⁸ Thus, dealumination via high- temperature calcination mainly occurred on Al of Si(OH)Al (Figs. 2(f, g)).' This is an assumption, not an evidence

Reply 7: Thank the careful check from reviewer 3. The evidence of dealumination has been provided in Fig. 2e. The two sentences are consecutive “Considering the mechanism of acid amount reduction by calcination (Fig. 2e), the schematic diagram of the relationship between Cu and BAS was depicted in Figs. 2(f, g) and Supplementary Fig. 27. It is worth pointing out that cations such as Cu²⁺ and Na⁺ on the extra framework of zeolite can stabilize Al in the framework.³⁸ Thus, dealumination via high-temperature calcination mainly occurred on Al of Si(OH)Al (Figs. 2(f, g)).” What we want to state is that for Cu/AEI zeolites, the dealumination occurred on the Si(OH)Al, not Si(OCu)Al, since Cu on Si(OCu)Al can stabilize Al on Si(OCu)Al.

REVIEWERS' COMMENTS

Reviewer #1 (Remarks to the Author):

I am happy with the changes made.

Reviewer #3 (Remarks to the Author):

After modifications, this work might be accepted for the publication